# What Makes a "Good" Data Augmentation in Knowledge Distillation – A Statistical Perspective

**Huan Wang**[1,2,†]    **Suhas Lohit**[2,*]    **Mike Jones**[2]    **Yun Fu**[1]
[1]Northeastern University, Boston, MA    [2]MERL, Cambridge, MA
Project: http://huanwang.tech/Good-DA-in-KD

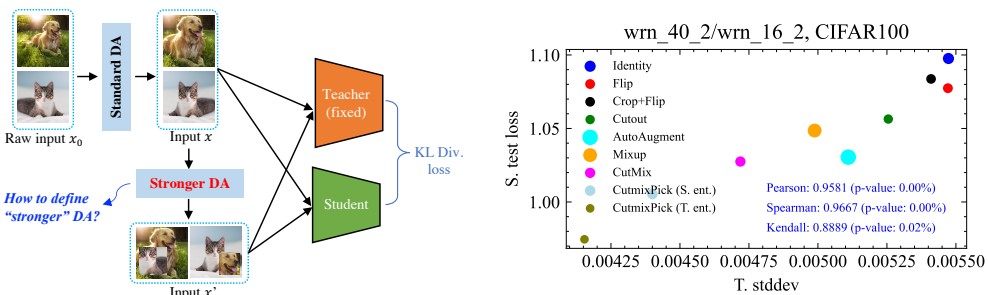

(a) Apply additional stronger DA in KD     (b) S. test loss *vs.* T. stddev with different DA schemes

Figure 1: **(a)** Illustration of applying a stronger data augmentation (DA) in addition to the standard DA (random crop and flip) in knowledge distillation (KD). We ask: *What makes a "good" DA when it is applied to KD in the manner of (a)?* **(b)** We present a proven proposition (Proposition 3.1) to answer this question rigorously, along with a practical metric to evaluate the "goodness" of a DA. The proposed metric is called *stddev of teacher's mean probability* (shorted as T. stddev). As seen in (b), there is a *strong* positive correlation (p-value < 5% is typically considered statistically significant) between the student's test loss (S. test loss) and T. stddev, showing that T. stddev well captures the "goodness" of different DA schemes in KD. The most striking fact from this plot may be: T. stddev is *purely* calculated with the teacher (no any student used) while it can "predict" the relative order of the *student*'s performance, implying the "goodness" of DA in KD probably is student-invariant.

## Abstract

Knowledge distillation (KD) is a general neural network training approach that uses a teacher model to guide the student model. Existing works mainly study KD from the network output side (*e.g.*, trying to design a better KD loss function), while few have attempted to understand it from the input side. Especially, its interplay with data augmentation (DA) has not been well understood. In this paper, we ask: Why do some DA schemes (*e.g.*, CutMix) inherently perform much better than others in KD? What makes a "good" DA in KD? Our investigation from a statistical perspective suggests that **a good DA scheme should reduce the covariance of the teacher-student cross-entropy**. A practical metric, *the stddev of teacher's mean probability* (T. stddev), is further presented and well justified empirically. Besides the theoretical understanding, we also introduce a new entropy-based data-mixing DA scheme, *CutMixPick*, to further enhance CutMix. Extensive empirical studies support our claims and demonstrate how we can harvest considerable performance gains simply by using a better DA scheme in knowledge distillation.

---

[†]This paper originates from Huan's summer internship work at MERL.
[*]Corresponding author: `slohit@merl.com`

36th Conference on Neural Information Processing Systems (NeurIPS 2022).

# 1 Introduction

Deep neural networks (DNNs) are the de facto methodology in many artificial intelligence areas nowadays [23, 35]. How to effectively train a deep network has been a central topic for decades. In the past several years, efforts have mainly focused on better architecture design (*e.g.*, batch normalization [19], residual blocks [13], dense connections [18]) and better loss functions (*e.g.*, label smoothing [41, 28], contrastive loss [17], large-margin softmax [24]) than the standard cross-entropy (CE) loss. Knowledge distillation (KD) [16] is a training method that falls into the second group. In KD, a stronger network – called teacher – is introduced to guide the learning of the original network – called student – by minimizing the discrepancy between the representations of the two networks,

$$\mathcal{L}_{KD} = (1 - \alpha)\mathcal{L}_{CE}(y, \mathbf{p}^{(s)}) + \alpha\tau^2 \mathcal{D}_{KL}(\mathbf{p}^{(t)}/\tau, \mathbf{p}^{(s)}/\tau), \tag{1}$$

where $\mathcal{D}_{KL}$ represents KL divergence [22]; $\alpha \in (0, 1)$ is a factor to balance the two loss terms; $\mathcal{L}_{CE}$ denotes the cross-entropy loss; $y$ is the one-hot label and $\mathbf{p}^{(t)}, \mathbf{p}^{(s)}$ stand for the teacher's and student's output probabilities over the classes; $\tau$ is a temperature constant [16] to smooth predicted probabilities. KD allows us to train smaller, more efficient neural networks without compromising on accuracy, which facilitates deploying deep learning in resource constrained environments (*e.g.*, on mobile devices). KD has found plenty of applications in many tasks [6, 46, 12, 20, 47].

Most existing KD methods have attempted to improve it by proposing better KD loss functions applied at the network outputs [32, 29, 43]. Few works have considered KD from *the input side*. Especially, the interplay between KD and data augmentation (DA) [38] has not been well understood so far (note, by DA here, we mean the conventional DA concept: generating a new input by *transforming one or multiple inputs*. A broader scope of DA may involve the *neural network*, *e.g.*, dropout can be seen as a kind of DA [3]. We do not consider this type of DA in this paper due to the limited length).

In this work, we ask: *What makes a "good" data augmentation in knowledge distillation*? A clear answer to this question has many benefits. First, theoretically, it can help us towards a better understanding about how data augmentation plays a role in KD. Second, practically, it can bring us considerable performance gain – in Fig. 2, we show test error rates using the standard CE loss (no teacher) *vs.* using KD loss (with a teacher). As seen, a stronger DA can lower the test error rate and admit more training iterations without overfitting in KD. It is pretty obvious to see that "Flip+Crop" is stronger than "Flip" alone in Fig. 2. However, for other DA schemes, such as Mixup [51] *vs.* AutoAugment [7], which one is stronger? Not very clear. We thus desire a principled way (*e.g.*, a concrete metric) to make the vague concept "stronger" exact. Presenting such a theoretically sound metric and empirically validating its effectiveness is the goal of this paper.

Intuitively, a good DA should enrich the input data and expose more knowledge of the teacher so that the student can generalize better. We formalize this idea from a statistical learning perspective. Specifically, we will show a good DA scheme is defined by a lower variance (or covariance) of the *teacher's mean output probability* over different input samples, which ultimately leads to a lower generalization gap for the student. The proposed theory is well justified by our extensive empirical studies on CIFAR100 and Tiny ImageNet datasets with various pairs. The proposed theory well explains why Cut-Mix is better than other alternatives (such as Mixup [51], AutoAugment [7]) in KD.

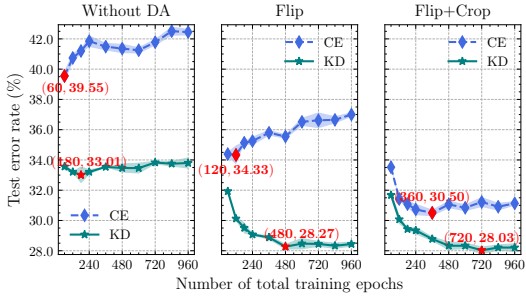

Figure 2: Test error rate of resnet20 on CIFAR100 when trained for different numbers of epochs with (KD) and without (CE) knowledge distillation (the teacher is resnet56 for KD). Each result is obtained by averaging 3 random runs (shaded area indicates the stddev). "Flip": random horizontal flip; "Crop": random crop. The optimal number of training epochs and its test loss are highlighted in red.

In addition to the new theoretical results, we also propose an entropy-based data picking scheme to select more informative samples for KD, which can deliver even lower variance of the teacher's mean probability as well as lower generalization error of the student.

We make the following contributions in this paper:

- We present a proven proposition (Proposition 3.1) that precisely answers what defines a better DA in KD: Given a fixed teacher, a better DA is the one that gives a lower variance of the teacher's mean probability.

- The proposition is well justified empirically on standard image classification datasets with many teacher-student pairs.

- An entropy-based data picking scheme is introduced to further reduce the variance of the teacher's mean probability, which can further advance CutMix, the prior state-of-the-art DA approach among those evaluated in this paper.

- Empirically, we show how the presented theory can benefit in practice – we can simply enhance the existing KD methods by using a stronger DA and prolonged training iterations.

## 2   Related Work

**Knowledge Distillation (KD)**. The general idea of knowledge distillation is to guide the training of a student model through a (stronger) teacher model (or an ensemble of models). It was pioneered by Buciluǎ *et al.* [4] and later refined by Hinton *et al.* [16], who coined the term. Since its debut, KD has seen extensive application in vision and language tasks [6, 14, 46, 20, 47]. Many variants have been proposed regarding the central question in KD, that is, how to define the *knowledge* transferred from the teacher to the student. Examples of such knowledge definitions include feature distance [33], feature map attention [50], feature distribution [30], activation boundary [15], inter-sample distance relationship [29, 32, 25, 44], and mutual information [43]. Several works [28, 37, 5] investigate the connection between label smoothing and knowledge distillation. Another line of works (*e.g.*, [42]) attempts to understand KD more theoretically. Over the past several years, the progress has been made primarily for intermediate feature maps and network outputs (*i.e.*, through a better loss function). In contrast, our goal is to improve the KD performance at the *input end* with the help of data augmentation. We will show this path is as effective and also has much potential for future research.

**Data Augmentation (DA)**. Deep neural networks are prone to overfitting, *i.e.*, building input-target mappings using undesirable or irrelevant features (like noise) in the data. Data augmentation is a prevailing technique to curb overfitting [38]. In classification tasks, data augmentation aims to explicitly provide data with *label-invariant transformations* (such as random crop, horizontal flip, color jittering, Cutout [11]) during training so that the model can learn representations robust to those nuisance factors. Recently, more advanced data augmentation methods were proposed, which not only transform the input, but also transform the target. For example, Mixup [51] linearly mixes two images with the labels mixed by the same linear interpolation. Manifold Mixup [45] is similar to Mixup but conducts the mix operation in the feature level instead of pixel level; CutMix [48] pastes a patch cut from an image onto another image with the label decided by the area ratio of the two parts. AutoAugment [7] is a strong DA method that finds an optimal augmentation policy from a large search space via reinforcement learning. When both the input and target are transformed simultaneously, the key is to maintain a *semantic correspondence* between the new input and new target. Unlike these methods, which focus on general classification using the cross-entropy loss, our work investigates the interplay between data augmentation and knowledge distillation loss and proposes new data augmentation specifically for knowledge distillation.

Some recent KD works also involve the utilization of DA in KD, such as [2, 8]. Especially, [2] also employs Mixup and prolonged training to enhance the student performance in KD, akin to ours. Yet it is worthwhile to note that our paper is *substantially different* from theirs, in that we are seeking the *theoretical* reason explaining how to define a better DA in KD to deliver better performance; these works mainly investigate in an empirical fashion, with no theoretical results presented. Meanwhile, our work is not limited to one specific DA (see Sec. 5). We target a *general theoretical* understanding which can apply to a broad scope of DA schemes (fortunately, as our experiments show, the proposed theory and the derived DA "goodness" measure indeed capture it). The fact that [2] utilizes Mixup [51] and prolonged training to deliver 82.8% top-1 accuracy with resnet50 [13] on ImageNet [10] can be a direct proof that the principle proposed in this work can bring us very promising practical benefits.

One recent work [9] also conducts empirical studies of the impact of DA on KD. They first apply DA (*e.g.*, Mixup/CutMix) to the teacher training then conduct the KD step as usual (no extra DA in this step). Our investigation is the *exact opposite* to their setup: We train the teacher as usual (no Mixup/CutMix), then in the KD step we employ a more advanced DA (*e.g.*, Mixup/CutMix).

Interestingly, they conclude that the teacher trained with Mixup/CutMix *hurts* the student's generalization ability, while we consistently see student performance boost via a stronger DA. Another recent work [40] utilizes KD to re-label mixed samples in Mixup to fix the inaccurate labelling problem of Mixup. Their work shows that KD can be used to make Mixup more generally useful, which is orthogonal to the topic of this work.

The work of [27] presents a statistical perspective to understand how the softened probabilities in KD are better than the one-hot hard labels. Our work is inspired by their *Bayes teacher* notion. This said, our work is different from theirs in that we focus on explaining how data augmentation plays a role in KD and answer what characterizes a good DA, while they attempt to answer why the knowledge distillation loss is better than the standard cross-entropy loss.

## 3 Theoretical Investigation

### 3.1 Prerequisites: Multi-Class Classification with KD

Given a training set $S = \{(x_n, y_n)\}_{n=1}^N \sim \mathscr{D}^N$, where $\mathscr{D}$ is the joint distribution for input-output random variable pair $(x, y)$, the goal in multi-class classification is to pin down a predictor $\mathbf{f} : \mathcal{X} \to \mathbb{R}^C$ from a hypothesis class $\mathcal{H}$, where $\mathcal{X}$ is the input space and $C$ refers to the number of classes. The predictor $\mathbf{f}$ is supposed to minimize the *true risk*

$$R_{\mathscr{D}}(\mathbf{f}) \overset{\text{def}}{=} \underset{(x,y)\sim\mathscr{D}}{\mathrm{E}} \left[ L(y, \mathbf{f}(x)) \right], \tag{2}$$

where $L$ stands for the loss objective function (*e.g.*, cross-entropy); the subscript $\mathscr{D}$ of $R_{\mathscr{D}}$ is to emphasize that the true risk is defined on the data distribution. The true risk is approximated in practice on a separate test set.

For training, the predictor aims to minimize the *empirical risk* defined on the training sequence $S$:

$$R_S(\mathbf{f}) \overset{\text{def}}{=} -\frac{1}{N} \sum_{n=1}^N \mathbf{e}_{y_n}^\top \log(\mathbf{f}(x_n)), \tag{3}$$

where $\mathbf{e}_y \in \{0, 1\}^C$ is a one-hot vector indicating the label $y \in [C] = \{1, 2, \cdots, C\}$. The subscript $S$ of $R_S$ is to emphasize the empirical risk is defined on the finite sampled data points.

In the context of KD, the one-hot hard target vector $\mathbf{e}_y$ is replaced with a probability vector $\mathbf{p}^{(t)}(x) \in \mathbb{R}_+^C$, giving us the *empirical distilled risk* of $\mathbf{f}$:

$$\hat{R}_S(\mathbf{f}) \overset{\text{def}}{=} -\frac{1}{N} \sum_{n=1}^N \mathbf{p}^{(t)}(x_n)^\top \log(\mathbf{f}(x_n)), \tag{4}$$

where the super-script $t$ indicates the fixed *teacher* model. The theory concerning why Eq. (4) is better than Eq. (3) has been established in [27]. Interested readers may refer to their paper for more details. Next, we look into how the data augmentation plays a role in KD.

### 3.2 What Makes a "Good" DA in KD?

Intuitively, a better DA should provide more information, *i.e.*, expose more knowledge of the teacher so that the student can absorb more and thus generalize better. We make this idea rigorous as follows.

**Proposition 3.1.** *Given a bounded loss function and a fixed teacher model with the empirical distilled risk defined in Eq. (4), for any predictor $\mathbf{f}$, consider two sampled sequences $S_1 \in \mathscr{D}^N$ and $S_2 \in \mathscr{D}^N$, they are made up of $N$ elements sampled from the same distribution $\mathscr{D}$, **while not i.i.d.** (especially when data augmentation is employed). If the elements in $S_1$ present a larger correlation than those in $S_2$, then the student's generalization gap trained on $S_1$ will be greater than that trained on $S_2$:*

$$\underset{S_1\sim\mathscr{D}^N}{\mathrm{E}} \left[ (\hat{R}_{S_1}(\mathbf{f}) - R_{\mathscr{D}}(\mathbf{f}))^2 \right] > \underset{S_2\sim\mathscr{D}^N}{\mathrm{E}} \left[ (\hat{R}_{S_2}(\mathbf{f}) - R_{\mathscr{D}}(\mathbf{f}))^2 \right]. \tag{5}$$

*Proof.* Let $\Delta = \hat{R}_S(\mathbf{f}) - R_{\mathscr{D}}(\mathbf{f})$. By the definition of variance, $\mathrm{E}_S[\Delta^2] = \mathrm{Var}_S[\Delta] + (\mathrm{E}_S[\Delta])^2$. To make the notation clearer, we define $q(x_i) = -\mathbf{p}^{(t)}(x_i)^\top \log(\mathbf{f}(x_i))$.

(1) Since $R_{\mathscr{D}}(\mathbf{f})$ is a constant (albeit unknown),

$$\mathrm{E}_{S\sim\mathscr{D}^N}[\Delta] = \mathrm{E}_S[\hat{R}_S(\mathbf{f})] + \mathrm{Const} = \mathrm{E}_S[\frac{1}{N}\sum_{i=1}^N q(x_i)] + \mathrm{Const}$$

$$= \frac{1}{N}\sum_{i=1}^N \mathrm{E}_S[q(x_i)] + \mathrm{Const} = \frac{1}{N}\sum_{i=1}^N \mathrm{E}_{x_i}[q(x_i)] + \mathrm{Const} \tag{6}$$

$$= \frac{1}{N}\cdot N\cdot \mathrm{E}_x[q(x)] + \mathrm{Const} = \mathrm{E}_x[q(x)] + \mathrm{Const},$$

where the second last equation is because each element $x_i$ in $S$ is drawn from the same distribution $\mathscr{D}$. From the RHS of the last equation, we can clearly see that for $S_1$, $S_2$, $\mathrm{E}_S[\Delta]$ is the same.

(2) Then we consider $\mathrm{Var}_S[\Delta]$. Again, since $R_{\mathscr{D}}(\mathbf{f})$ is a constant, we only need to consider

$$\mathrm{Var}_S[\hat{R}_S(\mathbf{f})] = \mathrm{Var}_S[\frac{1}{N}\sum_{i=1}^N q(x_i)] = \frac{1}{N^2}\mathrm{Cov}_S[\sum_{j=1}^N q(x_j), \sum_{k=1}^N q(x_k)]$$

$$= \frac{1}{N^2}\Big(\sum_{i=1}^N \mathrm{Var}_{x_i}[q(x_i)] + 2\sum_{1\le j<k\le N}\mathrm{Cov}_S[q(x_j), q(x_k)]\Big) \tag{7}$$

$$= \frac{1}{N^2}\Big(N\cdot\mathrm{Var}_x[q(x)] + 2\sum_{1\le j<k\le N}\mathrm{Cov}_S[q(x_j), q(x_k)]\Big)$$

$$= \frac{1}{N}\mathrm{Var}_x[q(x)] + \frac{2}{N^2}\sum_{1\le j<k\le N}\mathrm{Cov}_S[q(x_j), q(x_k)],$$

where $\mathrm{Cov}[\cdot,\cdot]$ stands for covariance. From the last item in Eq. (7), we can see that the covariance part is different for different sampled $S$'s. If the samples in $S_1$ present a larger correlation than those in $S_2$, we will have $\mathrm{Var}_{S_1}[\Delta] > \mathrm{Var}_{S_2}[\Delta]$, which further leads to increased generalization gap for the student, *i.e.*, $\mathrm{E}_{S_1}[(R_{S_1}(\mathbf{f}) - R_{\mathscr{D}}(\mathbf{f}))^2] > \mathrm{E}_{S_2}[(R_{S_2}(\mathbf{f}) - R_{\mathscr{D}}(\mathbf{f}))^2]$. The proof is finished.  □

**Practical Use: Stddev of Teacher's Mean Probability (T. stddev).** Note in Proposition 3.1, the predictor $\mathbf{f}$ (*i.e.*, the student model) can be any one (not necessarily a converged model). Each student would have an order of different DA schemes regarding which is better "in its opinion". Presumably, different students will lead to different such orders. For practical use, we must pick a certain student as oracle to conduct the evaluation. Here, we can play a trick that can make it rather simple to use our theory – *assume* there is a student that performs *exactly the same* as the teacher (this assumption is not unpractical, since one straightforward example is to use the teacher as student). Then we have

$$\mathrm{Cov}[q(x_j), q(x_k)] = \mathrm{Cov}[\mathbf{p}^{(t)}(x_j)^\top \log(\mathbf{f}(x_j)), \mathbf{p}^{(t)}(x_k)^\top \log(\mathbf{f}(x_k))]$$
$$= \mathrm{Cov}[\mathbf{p}^{(t)}(x_j)^\top \log(\mathbf{p}^{(t)}(x_j)), \mathbf{p}^{(t)}(x_k)^\top \log(\mathbf{p}^{(t)}(x_k)). \tag{8}$$

In this case, we only need the covariance of the *teacher's* probability to measure the "goodness" of a certain DA technique, *no need for the student*. Despite not using any information of the student, the proposed metric turns out to correlate surprisingly well with the student's performance (see Fig. 4).

Based on Eq. (8), when using $S_1$ *vs.* $S_2$ as training sequence, the fundamental factor answering for the variance gap of $\mathrm{Var}_S[\Delta]$ boils down to the covariance in $\{\mathbf{p}^{(t)}(x_i)\}_{i=1}^N$. That is, a larger covariance in $\{\mathbf{p}^{(t)}(x_i)\}_{i=1}^N$ leads to a larger covariance in $\{\mathbf{p}^{(t)}(x_i)^\top \log(\mathbf{p}^{(t)}(x_i))\}_{i=1}^N$ in Eq. (8), which ultimately leads to a higher $\mathrm{Var}_S[\Delta]$ in Eq. (7).

The next step is to find a feasible way to estimate the covariance in $\{\mathbf{p}^{(t)}(x_i)\}_{i=1}^N$. Consider the *average* variable (denoted as $\mathbf{u}$ here) of several random variables $\{\mathbf{p}^{(t)}(x_k)\}_{k=1}^K$,

$$\mathbf{u} = \frac{1}{K}\sum_{x_k\in S^*}\mathbf{p}^{(t)}(x_k), \mathbf{u}\in\mathbb{R}_+^C. \tag{9}$$

Its variance (or equivalently, stddev) inherently takes into account the covariance among its addends. Therefore, we can use the variance of $\mathbf{u}$ as a proxy for the covariance in $\{\mathbf{p}^{(t)}(x_k)\}_{k=1}^K$:

$$\mathbf{m} = \mathrm{Var}_{S^*}(\mathbf{u}), \mathbf{m}\in\mathbb{R}_+^C, \quad \bar{m} = \frac{1}{C}\sum_{i\in[C]}(\mathbf{m}_i)^{\frac{1}{2}}, \bar{m}\in\mathbb{R}_+, \tag{10}$$

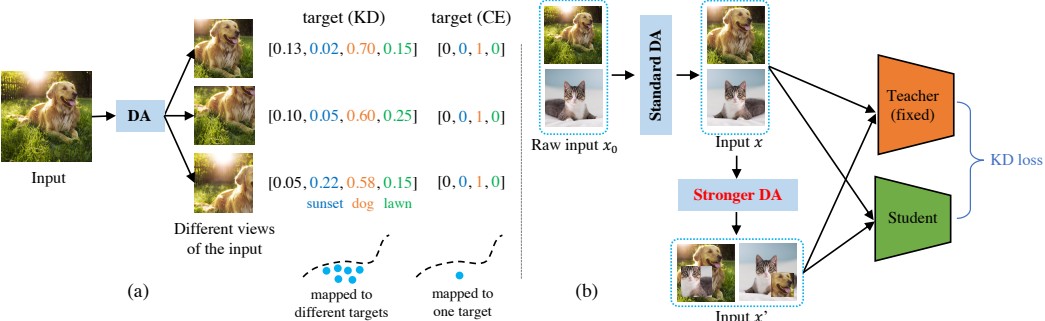

Figure 3: **(a)** Illustration of the difference of supervised target between the KD loss and cross-entropy (CE) loss. An input is transformed to different versions owing to data augmentation. KD loss can provide extra information to the student by mapping these views to different targets, while the CE loss cannot. This work attempts to answer what characterizes a "good" data augmentation scheme in distillation. **(b)** Illustration of adapting an existing DA approach to KD. The standard DA consists of random crop and horizontal flip. This training framework is employed to empirically verify our proposed metric to define a "stronger" data augmentation.

where $K$ is a pre-defined number of samples of set $S^*$ to realize the averaging effect (*e.g.*, $K = 640$ in our experiments for CIFAR100 and Tiny ImageNet – see Appendix Sec. **??** for a concrete calculation example). Note, $\mathbf{u}$ can be interpreted as the *teacher's mean probability* over $K$ input samples.

If there is a large covariance among $\{\mathbf{p}^{(t)}(x_k)\}_{k=1}^K$, $\mathbf{m}$ will be large (in an element-wise sense). Then the average of $\mathbf{m}$ over classes, *i.e.*, the $\bar{m}$ (we do this averaging simply because we desire a *scalar* metric), is a good indicator to capture such covariance. *We thus formally introduce $\bar{m}$, the (averaged) T. stddev, as the proposed metric to measure the quality of a data augmentation scheme.* A lower $\bar{m}$ implies a better DA by our definition. In the experiments, we will show this metric defined purely using the *teacher* can accurately characterize the generalization error of the *students* after distillation.

## 4 Evaluated Algorithms

### 4.1 Extend Existing DA Approaches for KD

Given an existing DA approach, this section explains how we adapt it properly to the case of KD.

Specifically, let $x_0$ denote the raw data, $x$ denote the transformed data by the standard augmentation (random crop and flip). Illustrated in Fig. 3(b), we will add the DA following $x$ to obtain $x'$. Unlike the common data augmentation where *only* the transformed input is fed into the network, we keep *both the input $x$ and $x'$* for the training (as such, the number of input examples is doubled). The consideration of keeping both inputs is to maintain the information path for the original input $x$ so that we can easily see how the added information path of $x'$ leads to a difference.

For $x$, its loss is still the original KD loss, consisting of the cross-entropy loss and the KL divergence (Eq. (1)). Of special note is that, for $x'$, its loss is *only* the KL divergence, *i.e.*, *we do not use the labels assigned by the DA algorithm* (for example, in the original Mixup and CutMix, they assign a linearly interpolated label to the augmented sample) because these labels can actually be *wrong* (see Appendix Sec. **??** for concrete examples on ImageNet). In fact, not using the hard label has another bonus. A dataset augmentation scheme which employs CE loss has to provide corresponding labels as supervisory information. In order to maintain the semantic correspondence, it cannot admit very extreme transformations for data augmentation. In contrast, in the Mixup/CutMix+KD setting described above, the data augmentation scheme need not worry about the labels as they are assigned *by the teacher* – the recent work [2] refers this kind of utilization of DA as *function matching* of the teacher. As a result, it can admit a *broader* set of transformations to expose the teacher's knowledge more completely. This reflects that data augmentation in KD has more freedom than in CE.

Among all the data augmentation techniques evaluated in this paper, we will show *CutMix works best*. The fundamental reason for its success, as we will show, is that it achieves a much lower variance of the teacher's mean probability, implying it produces more diverse data than its counterparts.

## 4.2 CutMixPick: Enhancing CutMix with Entropy-Based Data Picking

In this section, we propose a data picking scheme to further reduce the variance of the teacher's mean probability. The idea is partly inspired by *active learning* [36]. In active learning, the learner enjoys the freedom to query the data instances to be labeled for training by an oracle (*i.e.*, the teacher in our case) [36]. Since the augmented data can vary in their quality, we can introduce a certain criterion to pick the more valuable data for the student.

Intuitively, we regard a sample with more *information* is of higher quality. Therefore, we take *Shannon's entropy* of the teacher's output probability as a natural measure to select samples,

$$H(\mathbf{p}^{(t)}(x)) = -\mathbf{p}^{(t)}(x)^\top \log(\mathbf{p}^{(t)}(x)). \tag{11}$$

Note this formula is in exactly the same form of Eq. (8). Empirically, we will also show the data selected by this formula indeed results in lower $\bar{m}$ and better test loss for the student.

Concretely, given a batch of data, we first apply CutMix to obtain a bunch of augmented samples. Then sort all the augmented samples by Eq. (11) in ascending order and keep the top $r$ (a pre-defined percentage constant, $r = 0.5$ in our experiments) samples.

This simple technique can be rather effective according to our empirical study. Another seemingly potential alternative is to use the *student's* entropy as the picking metric. The intuition behind this is that a high-entropy sample in the view of the student can be regarded as a hard example, too. Learning with these hard examples may expand the student's knowledge by squeezing its blind spots. Despite this intuitively plausible explanation, in practice, we will show this scheme actually under-performs the former student-agnostic scheme in Eq. (11), which is a bit surprising.

## 5 Experimental Results

**Datasets and Networks**. We evaluate our method primarily on the CIFAR100 [21] and Tiny ImageNet* datasets. CIFAR100 has 100 object classes ($32\times32$ RGB images). Each class has 500 images for training and 100 images for testing. Tiny ImageNet is a small version of ImageNet [10] with 200 classes ($64\times64$ RGB images). Each class has 500 images for training, 50 for validation and 50 for testing. To thoroughly evaluate our methods, we benchmark them on various standard network architectures: vgg [39], resnet [13], wrn [49], MobileNetV2 [34], ShuffleV2 [26]. We will also include results on ImageNet100 (a randomly drawn 100-class subset of ImageNet) and ImageNet.

**Benchmark Methods**. In addition to the standard cross-entropy training and the original KD method [16], we also compare with the state-of-the-art distillation approach, *Contrastive Representation Distillation* (CRD) [43]. It is important to note that our method focuses on improving KD by using better *inputs*, while CRD improves KD at the *output* end (*i.e.*, a better loss function). Therefore, they are orthogonal and we will show they can be combined together to deliver even better results.

**Hyper-Parameter Settings**. The temperature $\tau$ of knowledge distillation is set to 4 following CRD [43]. Loss weight $\alpha = 0.9$ (Eq. (1)). For CIFAR100 and Tiny ImageNet, training batch size is 64; the original number of total training epochs is 240, with learning rate (LR) decayed at epoch 150, 180, and 210 by multiplier 0.1. The initial LR is 0.05. All these settings are *the same* as CRD [43] for fair comparison. Note, in our experiments we will present the results of more training iterations. If the number of total epochs is scaled by a factor $k$, the epochs after which learning rate is decayed is also be scaled by $k$. For example, if we train a network for CIFAR100 for 480 epochs ($k = 2$) in total, the learning rate will be decayed at epoch 300, 360, and 420. We use PyTorch [31] to conduct all our experiments. For CIFAR100, we adopt the pretrained teacher models from CRD† for fair comparison. For Tiny ImageNet and ImageNet100, we train our own teacher models. For ImageNet, we use torchvision models following CRD [43].

**Data Augmentation Schemes**. We investigate the following popular DA schemes:

- **Identity**: This augmentation scheme simply makes a copy of each batch data during training, which should be the *lower bound* of all DA schemes discussed here since it adds no new information.

- **Flip**: Random horizontal flip.

---

*https://tiny-imagenet.herokuapp.com/

†https://github.com/HobbitLong/RepDistiller

- **Flip+Crop**: Random horizontal flip and random crop. This is the standard DA extensively used in 2D image recognition task (such as on CIFAR and ImageNet datasets).
- **Cutout** [11]: Cutout occludes a small random patch of an image.
- **AutoAugment** [7]: AutoAugment is an ensemble of a collected DA schemes. The DA policy is automatically selected by reinforcement learning instead of manually.
- **Mixup** [51]: Mixup applies linear interpolation between two inputs and applies the same linear interpolation to their labels to make the new label for the augmented sample.
- **CutMix** [48]: CutMix cuts a small patch from a source image and pastes it to another source image. The resulted image is considered as a new input. The target for the new input is a linear interpolation from the two source labels.

## 5.1 Empirical Verification of Our Proposition

Before presenting results, one point worth mentioning is that, in this section we use *test loss* instead of accuracy as the measure of student's performance in KD, because **(1)** all the formulas in Sec. 3 are derived using numerical loss instead of accuracy; **(2)** more importantly, it is observed that accuracy can mismatch with loss – a model may achieve a better accuracy, meanwhile higher loss too [27], which we also observe several times in our experiments. Using accuracy as measure would prevent us from seeing the correlation between T. stddev and student's performance (see Appendix Fig. **??** for an example). Potential extension of our theory from loss to accuracy is left for future work.

In Fig. 4, we plot the scatters of S. test loss and T. stddev. The x/y-axis value of each data point is averaged by at least three random runs (see Tabs **??/??** and Tabs **??/??** for detailed numbers).

**(1)** In terms of T. stddev, there is a rough trend (*e.g.*, on the vgg13/vgg8 pair on CIFAR100): Identity < Flip < Flip+Crop < Cutout < AutoAugment < Mixup < CutMix. These inequalities are well-aligned with our intuition. *E.g.*, AutoAugment [7] includes Cutout [11] in its transformation pool, thus should be stronger than Cutout. This is faithfully reflected by the T. stddev on many pairs.

**(2)** Obviously, S. test loss poses a *clear positive correlation* with T. stddev. Per our theory, lower T. stddev should lead to better generalization risk for the student. This is generally well verified in these plots. We do see some minor counterexamples. Possible reasons are: *1)* The test loss is obtained on the test set with finite samples, which is only an approximation of the true risk defined on distribution; *2)* The teacher's mean probability is also evaluated on finite data. Despite them, the general picture from Fig. 4 still confirms the positive correlation between S. test loss and T. stddev. The correlation is actually *very significant* as the p-values indicate.

**(3)** Importantly, note we only need the teacher to define the quality of a certain DA in KD, *no need for the student*. This is more clear if we examine the results in Tabs. **??/??** and Tabs. **??/??**. Taking vgg13/vgg8 and vgg13/MobileNetV2 as an example, the students are starkly different but both the student's performance correlates well with T. stddev. This observation implies that *the "goodness" of DA in KD is probably student-invariant*. This, notably, is a great advantage in practice since we can decide which DA should be used for the best performance simply using a formula (Eq. (10)) with a few network forwards, without having to train the students physically.

## 5.2 Boosting KD with Stronger DA

In this section, we present more results to show how the proposed theory can benefit in practice – we can harvest considerable performance gain simply by using a stronger DA in KD.

**Prolonged Training**. Notably, a stronger DA produces more diverse data, implying more information. Intuitively, it should take a student model *more training iterations* (if the batch size does not change) to fully absorb the excessive information. That is, a stronger DA conceivably takes *more* training iterations to fully exhibit its potential. This intuition is confirmed in Fig. 2 (and also two more pairs wrn_40_2/wrn_16_2 and vgg13/vgg8 in the Appendix). Thus, in our experiments, we will also report results with prolonged training iterations for maximized performance.

**Results on CIFAR100**. The results on CIFAR100 dataset are shown in Tab. 1.

**(1)** Comparing the row "KD+CutMix" to "KD", we see CutMix improves the student accuracies on *all* the pairs. On the pair resnet32x4/ShuffleV2, the improvement is very significant (more than 1 percentage point). **(2)** Comparing the row "KD+CutMixPick" to "KD+CutMix", we see 6/7 pairs

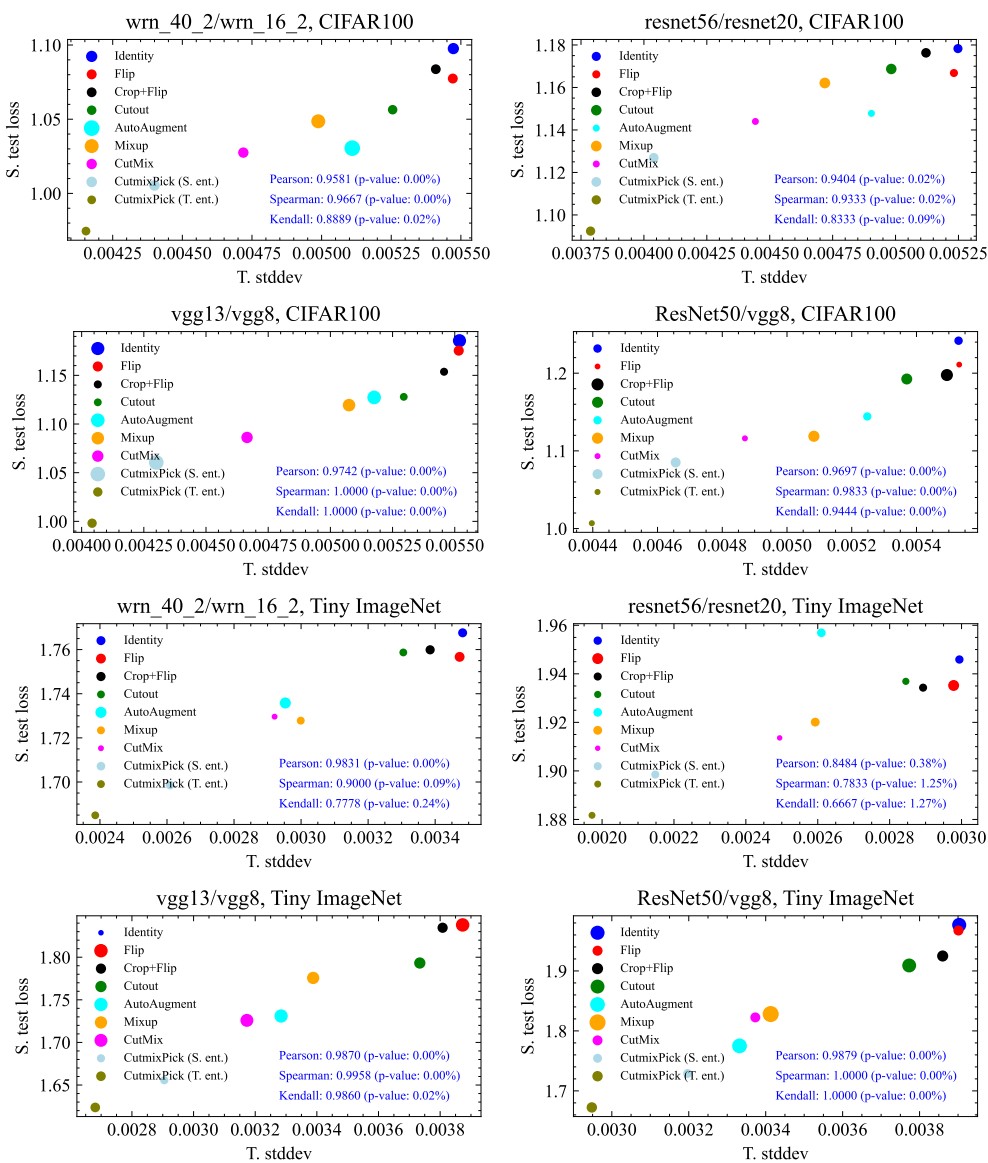

Figure 4: Scatter plots of T. stddev *vs.* S. test loss of different pairs on CIFAR100 and Tiny ImageNet. The detailed numbers are deferred to Appendix (Tab. **??**, Tab. **??**, Tab. **??**, Tab. **??**).

are improved further, showing the proposed data picking scheme works in most cases. **(3)** Finally, "KD+CutMixPick" scheme can be combined with more training iterations (960 epochs), which delivers even higher accuracies. **(4)** If comparing our best results (KD+CutMixPick$_{960}$) with those of CRD (though this is not an apples-to-apples comparison since the two methods focus on different aspects to improve KD), we can see our approach outperforms CRD on 6/7 pairs.

In the last two rows of Tab. 1, when CRD [43] is armed with our proposed "CutMixPick" and more training iterations, its results can be further advanced *consistently*. This demonstrates that our method is general and can readily work with those methods focusing on better KD loss functions.

**Results on Tiny ImageNet**. We also evaluate CutMix and CutMixPick on a more challenging dataset, Tiny ImageNet. Similar to the case on CIFAR100, we have results on different teacher-student pairs, shown in Tab. 2. For prolonged training, we train for 480 epochs instead of 960 to save time. Most claims on the CIFAR100 dataset are also validated here: **(1)** "KD+CutMix" is better than KD, which is verified on *all* pairs. **(2)** "KD+CutMixPick" is better than "KD+CutMix", verified on 6/7 pairs. The exception pair is resnet56/resnet20, where adding data picking decreases the accuracy slightly by 0.11%. **(3)** When "KD+CutMixPick" is trained for prolonged iterations, the students perform best.

Table 1: Student test accuracy on **CIFAR100**. Each result is obtained by 3 random runs, mean (std) accuracy reported. The best results are in **bold** and second best underlined. The subscript 960 means the total number of training epochs (default: 240).

| Teacher
Student | wrn_40_2
wrn_16_2 | resnet56
resnet20 | resnet32x4
resnet8x4 | vgg13
vgg8 | vgg13
MobileNetV2 | ResNet50
vgg8 | resnet32x4
ShuffleV2 |
|---|---|---|---|---|---|---|---|
| Teacher Acc.
Student Acc. | 75.61
73.26 | 72.34
69.06 | 79.42
72.50 | 74.64
70.36 | 74.64
64.60 | 79.34
70.36 | 79.42
71.82 |
| KD [16] | 74.92 (0.28) | 70.66 (0.24) | 73.33 (0.25) | 72.98 (0.19) | 67.37 (0.32) | 73.81 (0.13) | 74.45 (0.27) |
| KD+CutMix | 75.34 (0.19) | 70.77 (0.17) | 74.91 (0.20) | 74.16 (0.18) | 68.79 (0.35) | 74.85 (0.23) | 76.61 (0.18) |
| **KD+CutMixPick** | 75.59 (0.22) | 70.99 (0.20) | 74.78 (0.35) | 74.43 (0.20) | 69.49 (0.32) | 74.95 (0.18) | 76.90 (0.25) |
| KD$_{960}$ [16] | 75.68 (0.12) | **71.79** (0.29) | 73.14 (0.06) | 74.00 (0.34) | 68.77 (0.05) | 74.04 (0.25) | 74.64 (0.30) |
| **KD+CutMixPick$_{960}$** | **76.41** (0.10) | 71.66 (0.15) | **75.12** (0.18) | **75.00** (0.17) | **70.47** (0.12) | **76.13** (0.16) | **77.90** (0.30) |
| CRD [43] | 75.64 (0.21) | 71.63 (0.15) | 75.46 (0.25) | 74.29 (0.12) | 69.94 (0.05) | 74.58 (0.27) | 76.05 (0.09) |
| CRD+**CutMixPick** | 75.96 (0.27) | 71.41 (0.26) | **76.11** (0.53) | 74.65 (0.12) | 69.95 (0.22) | 75.35 (0.22) | 76.93 (0.11) |
| CRD+**CutMixPick$_{960}$** | **76.61** (0.01) | **72.40** (0.20) | 75.96 (0.29) | **75.41** (0.10) | **70.84** (0.05) | **76.20** (0.22) | **78.51** (0.27) |

Table 2: Student test accuracy on **Tiny ImageNet**. The subscript 480 means the total number of training epochs (default: 240).

| Teacher
Student | wrn_40_2
wrn_16_2 | resnet56
resnet20 | resnet32x4
resnet8x4 | vgg13
vgg8 | vgg13
MobileNetV2 | ResNet50
vgg8 | resnet32x4
ShuffleV2 |
|---|---|---|---|---|---|---|---|
| Teacher Acc.
Student Acc. | 61.28
58.23 | 58.37
52.53 | 64.41
55.41 | 62.59
56.67 | 62.59
58.20 | 68.20
56.67 | 64.41
62.07 |
| KD [16] | 58.65 (0.09) | 53.58 (0.18) | 55.67 (0.09) | 61.48 (0.36) | 59.28 (0.13) | 60.39 (0.16) | 66.34 (0.11) |
| KD+CutMix | 59.06 (0.18) | 53.77 (0.33) | 56.41 (0.04) | 62.17 (0.11) | 60.48 (0.30) | 61.12 (0.18) | 67.01 (0.30) |
| **KD+CutMixPick** | 59.22 (0.05) | 53.66 (0.05) | 56.82 (0.23) | 62.32 (0.18) | 60.53 (0.18) | 61.40 (0.26) | 67.08 (0.13) |
| KD$_{480}$ [16] | 59.20 (0.30) | 54.23 (0.24) | 55.49 (0.11) | 61.72 (0.10) | 59.27 (0.08) | 60.10 (0.30) | 65.81 (0.11) |
| **KD+CutMixPick$_{480}$** | **60.07** (0.04) | **54.25** (0.07) | **57.54** (0.23) | **62.60** (0.25) | **60.66** (0.15) | **61.95** (0.14) | **67.35** (0.21) |
| CRD [43] | 60.79 (0.24) | 55.34 (0.02) | 59.28 (0.13) | 62.92 (0.31) | 62.38 (0.19) | 62.03 (0.16) | 67.33 (0.13) |
| CRD+**CutMixPick** | 60.72 (0.09) | 54.99 (0.16) | 59.65 (0.24) | 63.39 (0.10) | 62.54 (0.22) | **62.85** (0.18) | 67.64 (0.18) |
| CRD+**CutMixPick$_{480}$** | **60.99** (0.33) | **55.68** (0.22) | **60.13** (0.13) | **63.60** (0.20) | **62.79** (0.03) | 62.60 (0.17) | **67.70** (0.35) |

We further evaluate our DA methods equipped with CRD [43], shown in the last two rows of Tab. 2. Our "CutMixPick" method further advances the prior SOTA on 5 pairs. When CRD+CutMixPick is trained for 480 epochs (instead of 240), further improvement can be observed on 6 of 7 pairs.

**Apply DA to More KD Methods**. Notably, we achieve the above performance boosting simply using the original KD loss [16], *with no bells and whistles*. This justifies one of our motivations in this paper, *i.e.*, existing KD methods [32, 29, 43] mainly improve KD at the network *output* side via better loss functions, while we propose to improve KD at the *input* side and show this path is just as promising. Actually, this performance boosting effect is *generic* – we also applied CutMix to another 5 top-performing KD methods on CIFAR100: AT [50], CC [32], SP [44], PKT [30], and VID [1]. *All* the pairs see accuracy gains; half of them are even improved by more than 1% point.

**Results on ImageNet100 and ImageNet**. These results are deferred to Appendix **??**. In general, we observe that the correlation between T. stddev and S. test loss become weaker on ImageNet100 and ImageNet. This is because these two datasets are inherently harder than CIFAR100 and Tiny ImageNet. Nevertheless, the p-value of the correlation is still below 5% on ImageNet100, *i.e.*, still statistically significant, suggesting our theory can generalize to large-resolution ($224 \times 224$) datasets.

## 6 Conclusion

In this paper, we attempt to precisely answer what makes a good data augmentation in knowledge distillation. By analyzing the generalization gap of the student under different sampling schemes, we reach the conclusion that a good data augmentation scheme should reduce the variance of the cross-entropy (*i.e.*, the distilled risk) between the teacher and student. Based on this, we propose a new metric, the stddev of the teacher's mean probability (T. stddev), as a feasible measure of the quality of data augmentation techniques. Empirical studies with various teacher-student pairs confirm the efficacy of the proposed metric for data augmentation quality in KD. Besides the theoretical understanding, we also develop an entropy-based data picking scheme to further enhance the prior best augmentation scheme (CutMix) in KD. Finally, we show how we can obtain considerable KD performance gains simply by using a stronger DA guided by the proposed theory.

## Acknowledgments and Disclosure of Funding

We thank the anonymous NeurIPS reviewers for giving us very helpful suggestions to improve this paper!

This work originates from Huan's internship at MERL, and is finished eventually after he returns to Northeastern University as a research assistant. This work is thus fully supported by MERL and Northeastern University. There is no third-party funding or support in any form. There are no competing interests to disclose.

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
