## A  Appendix

### A.1  More Results

**Detailed numerical results of S. test loss and T. stddev on CIFAR100 and Tiny ImageNet**. See Tab. 3, Tab. 4, Tab. 5, Tab. 6. These tables are the numerical results that we use to plot Fig. 4.

Table 3: Student test loss (S. test loss) and the stddev of teacher's mean probability (T. stddev, $\times 10^{-3}$) comparison on **CIFAR100** when using different DA schemes.

| Teacher
Student
Metric | wrn_40_2
/
T. stddev | wrn_40_2
wrn_16_2
S. test loss | wrn_40_2
vgg8
S. test loss | resnet56
/
T. stddev | resnet56
resnet20
S. test loss | resnet56
ShuffleV2
S. test loss |
|---|---|---|---|---|---|---|
| KD+Identity | $5.473_{\pm0.002}$ | $1.0976_{\pm0.0136}$ | $1.1830_{\pm0.0065}$ | $5.248_{\pm0.004}$ | $1.1783_{\pm0.0081}$ | $0.9785_{\pm0.0137}$ |
| KD+Flip | $5.471_{\pm0.004}$ | $1.0774_{\pm0.0101}$ | $1.1673_{\pm0.0060}$ | $5.232_{\pm0.003}$ | $1.1668_{\pm0.0060}$ | $0.9961_{\pm0.0072}$ |
| KD+Crop+Flip | $5.410_{\pm0.009}$ | $1.0837_{\pm0.0097}$ | $1.1446_{\pm0.0182}$ | $5.121_{\pm0.006}$ | $1.1763_{\pm0.0092}$ | $0.9736_{\pm0.0062}$ |
| KD+Cutout | $5.255_{\pm0.009}$ | $1.0564_{\pm0.0090}$ | $1.1306_{\pm0.0156}$ | $4.983_{\pm0.012}$ | $1.1687_{\pm0.0115}$ | $0.9541_{\pm0.0088}$ |
| KD+AutoAugment | $5.110_{\pm0.004}$ | $1.0305_{\pm0.0290}$ | $1.1102_{\pm0.0016}$ | $4.904_{\pm0.007}$ | $1.1478_{\pm0.0041}$ | $0.9355_{\pm0.0099}$ |
| KD+Mixup | $4.988_{\pm0.012}$ | $1.0486_{\pm0.0225}$ | $1.0917_{\pm0.0044}$ | $4.719_{\pm0.002}$ | $1.1621_{\pm0.0121}$ | $0.9703_{\pm0.0060}$ |
| KD+CutMix | $4.719_{\pm0.005}$ | $1.0275_{\pm0.0112}$ | $1.0657_{\pm0.0145}$ | $4.443_{\pm0.002}$ | $1.1440_{\pm0.0040}$ | $0.9348_{\pm0.0107}$ |
| KD+CutMixPick (S. ent.) | $4.400_{\pm0.007}$ | $1.0054_{\pm0.0118}$ | $1.0471_{\pm0.0111}$ | $4.039_{\pm0.004}$ | $1.1269_{\pm0.0095}$ | $0.9339_{\pm0.0070}$ |
| KD+CutMixPick (T. ent.) | $4.154_{\pm0.005}$ | $0.9746_{\pm0.0073}$ | $0.9928_{\pm0.0061}$ | $3.788_{\pm0.004}$ | $1.0924_{\pm0.0085}$ | $0.9038_{\pm0.0148}$ |

Table 4: **Continued**: Student test loss (S. test loss) and the stddev of teacher's mean probability (T. stddev, $\times 10^{-3}$) comparison on **CIFAR100** when using different DA schemes.

| Teacher
Student
Metric | vgg13
/
T. stddev | vgg13
vgg8
S. test loss | vgg13
MobileNetV2
S. test loss | resnet32x4
/
T. stddev | resnet32x4
resnet8x4
S. test loss | resnet32x4
ShuffleV2
S. test loss | ResNet50
/
T. stddev | ResNet50
vgg8
S. test loss |
|---|---|---|---|---|---|---|---|---|
| KD+Identity | $5.519_{\pm0.006}$ | $1.1856_{\pm0.0196}$ | $1.6021_{\pm0.0366}$ | $5.524_{\pm0.015}$ | $1.0973_{\pm0.0134}$ | $1.3381_{\pm0.0072}$ | $5.530_{\pm0.005}$ | $1.2418_{\pm0.0067}$ |
| KD+Flip | $5.516_{\pm0.002}$ | $1.1754_{\pm0.0105}$ | $1.5779_{\pm0.0076}$ | $5.530_{\pm0.009}$ | $1.0967_{\pm0.0038}$ | $1.3208_{\pm0.0172}$ | $5.532_{\pm0.002}$ | $1.2110_{\pm0.0024}$ |
| KD+Crop+Flip | $5.457_{\pm0.010}$ | $1.1537_{\pm0.0059}$ | $1.5244_{\pm0.0149}$ | $5.498_{\pm0.004}$ | $1.0685_{\pm0.0031}$ | $1.3091_{\pm0.0283}$ | $5.494_{\pm0.013}$ | $1.1976_{\pm0.0160}$ |
| KD+Cutout | $5.295_{\pm0.012}$ | $1.1279_{\pm0.0055}$ | $1.4963_{\pm0.0104}$ | $5.356_{\pm0.014}$ | $1.0627_{\pm0.0063}$ | $1.2528_{\pm0.0127}$ | $5.370_{\pm0.004}$ | $1.1925_{\pm0.0128}$ |
| KD+AutoAugment | $5.176_{\pm0.012}$ | $1.1273_{\pm0.0213}$ | $1.4137_{\pm0.0440}$ | $5.198_{\pm0.015}$ | $1.0289_{\pm0.0078}$ | $1.1353_{\pm0.0164}$ | $5.248_{\pm0.011}$ | $1.1443_{\pm0.0064}$ |
| KD+Mixup | $5.075_{\pm0.001}$ | $1.1194_{\pm0.0176}$ | $1.4009_{\pm0.0044}$ | $5.082_{\pm0.004}$ | $1.0182_{\pm0.0128}$ | $1.0840_{\pm0.0229}$ | $5.083_{\pm0.006}$ | $1.1188_{\pm0.0133}$ |
| KD+CutMix | $4.665_{\pm0.009}$ | $1.0862_{\pm0.0140}$ | $1.2846_{\pm0.0178}$ | $4.851_{\pm0.016}$ | $1.0166_{\pm0.0096}$ | $1.0544_{\pm0.0095}$ | $4.870_{\pm0.002}$ | $1.1161_{\pm0.0026}$ |
| KD+CutMixPick (S. ent.) | $4.300_{\pm0.016}$ | $1.0605_{\pm0.0249}$ | $1.2739_{\pm0.0064}$ | $4.590_{\pm0.006}$ | $0.9888_{\pm0.0015}$ | $1.0270_{\pm0.0047}$ | $4.656_{\pm0.012}$ | $1.0851_{\pm0.0101}$ |
| KD+CutMixPick (T. ent.) | $4.042_{\pm0.007}$ | $0.9981_{\pm0.0087}$ | $1.1876_{\pm0.0233}$ | $4.323_{\pm0.007}$ | $0.9485_{\pm0.0116}$ | $0.9570_{\pm0.0122}$ | $4.397_{\pm0.006}$ | $1.0069_{\pm0.0027}$ |

**ImageNet100 and ImageNet Results**. See Fig. 5, Tab. 7 and Tab. 8 for the results. In general, we find it is harder to verify the proposed proposition on these two more challenging datasets – As seen,

Table 5: Student test loss (S. test loss) and the stddev of teacher's mean probability (T. stddev, $\times 10^{-3}$) comparison on **Tiny ImageNet** when using different DA schemes.

| Teacher | wrn_40_2 | wrn_40_2 | wrn_40_2 | resnet56 | resnet56 | resnet56 |
|---|---|---|---|---|---|---|
| Student | / | wrn_16_2 | vgg8 | / | resnet20 | ShuffleV2 |
| Metric | T. stddev | S. test loss | S. test loss | T. stddev | S. test loss | S. test loss |
| KD+Identity | $3.482_{\pm0.002}$ | $1.7676_{\pm0.0079}$ | $1.7374_{\pm0.0111}$ | $2.994_{\pm0.001}$ | $1.9459_{\pm0.0065}$ | $1.5600_{\pm0.0077}$ |
| KD+Flip | $3.473_{\pm0.003}$ | $1.7567_{\pm0.0099}$ | $1.7322_{\pm0.0107}$ | $2.978_{\pm0.001}$ | $1.9352_{\pm0.0127}$ | $1.5629_{\pm0.0034}$ |
| KD+Crop+Flip | $3.385_{\pm0.004}$ | $1.7599_{\pm0.0083}$ | $1.7283_{\pm0.0083}$ | $2.893_{\pm0.000}$ | $1.9343_{\pm0.0060}$ | $1.5593_{\pm0.0048}$ |
| KD+Cutout | $3.305_{\pm0.002}$ | $1.7587_{\pm0.0055}$ | $1.7086_{\pm0.0041}$ | $2.845_{\pm0.003}$ | $1.9369_{\pm0.0044}$ | $1.5636_{\pm0.0052}$ |
| KD+AutoAugment | $2.953_{\pm0.002}$ | $1.7358_{\pm0.0131}$ | $1.7086_{\pm0.0386}$ | $2.610_{\pm0.002}$ | $1.9569_{\pm0.0071}$ | $1.5782_{\pm0.0085}$ |
| KD+Mixup | $2.999_{\pm0.002}$ | $1.7278_{\pm0.0057}$ | $1.7024_{\pm0.0066}$ | $2.593_{\pm0.004}$ | $1.9201_{\pm0.0075}$ | $1.5765_{\pm0.0036}$ |
| KD+CutMix | $2.921_{\pm0.005}$ | $1.7296_{\pm0.0025}$ | $1.6905_{\pm0.0033}$ | $2.494_{\pm0.005}$ | $1.9136_{\pm0.0020}$ | $1.5605_{\pm0.0024}$ |
| KD+CutMixPick (S. ent.) | $2.609_{\pm0.002}$ | $1.6986_{\pm0.0071}$ | $1.6524_{\pm0.0058}$ | $2.148_{\pm0.001}$ | $1.8985_{\pm0.0059}$ | $1.5580_{\pm0.0056}$ |
| KD+CutMixPick (T. ent.) | $2.386_{\pm0.002}$ | $1.6849_{\pm0.0055}$ | $1.6349_{\pm0.0060}$ | $1.972_{\pm0.003}$ | $1.8817_{\pm0.0040}$ | $1.5429_{\pm0.0060}$ |

Table 6: **Continued**: Student test loss (S. test loss) and the stddev of teacher's mean probability (T. stddev, $\times 10^{-3}$) comparison on **Tiny ImageNet** when using different DA schemes.

| Teacher | vgg13 | vgg13 | vgg13 | resnet32x4 | resnet32x4 | resnet32x4 | ResNet50 | ResNet50 |
|---|---|---|---|---|---|---|---|---|
| Student | / | vgg8 | MobileNetV2 | / | resnet8x4 | ShuffleV2 | / | vgg8 |
| Metric | T. stddev | S. test loss | S. test loss | T. stddev | S. test loss | S. test loss | T. stddev | S. test loss |
| KD+Identity | $3.873_{\pm0.001}$ | $1.8377_{\pm0.0021}$ | $2.0258_{\pm0.0211}$ | $3.847_{\pm0.004}$ | $1.9859_{\pm0.0108}$ | $1.7404_{\pm0.2064}$ | $3.903_{\pm0.001}$ | $1.9767_{\pm0.0230}$ |
| KD+Flip | $3.873_{\pm0.002}$ | $1.8379_{\pm0.0207}$ | $1.9946_{\pm0.0180}$ | $3.842_{\pm0.005}$ | $1.9531_{\pm0.0049}$ | $1.6092_{\pm0.0077}$ | $3.901_{\pm0.001}$ | $1.9674_{\pm0.0104}$ |
| KD+Crop+Flip | $3.808_{\pm0.004}$ | $1.8346_{\pm0.0108}$ | $1.9707_{\pm0.0089}$ | $3.784_{\pm0.002}$ | $1.9725_{\pm0.0092}$ | $1.6009_{\pm0.0111}$ | $3.860_{\pm0.003}$ | $1.9249_{\pm0.0132}$ |
| KD+Cutout | $3.734_{\pm0.066}$ | $1.7932_{\pm0.0137}$ | $1.9544_{\pm0.0101}$ | $3.688_{\pm0.001}$ | $1.9872_{\pm0.0047}$ | $1.5900_{\pm0.0092}$ | $3.773_{\pm0.003}$ | $1.9091_{\pm0.0223}$ |
| KD+AutoAugment | $3.284_{\pm0.003}$ | $1.7310_{\pm0.0200}$ | $1.7970_{\pm0.0445}$ | $3.269_{\pm0.003}$ | $1.9037_{\pm0.0248}$ | $1.4969_{\pm0.0159}$ | $3.332_{\pm0.005}$ | $1.7750_{\pm0.0255}$ |
| KD+Mixup | $3.388_{\pm0.002}$ | $1.7757_{\pm0.0168}$ | $1.8587_{\pm0.0141}$ | $3.326_{\pm0.002}$ | $1.9191_{\pm0.0136}$ | $1.5368_{\pm0.0108}$ | $3.413_{\pm0.001}$ | $1.8281_{\pm0.0309}$ |
| KD+CutMix | $3.173_{\pm0.001}$ | $1.7259_{\pm0.0189}$ | $1.8269_{\pm0.0423}$ | $3.275_{\pm0.003}$ | $1.9420_{\pm0.0082}$ | $1.5350_{\pm0.0083}$ | $3.373_{\pm0.004}$ | $1.8225_{\pm0.0101}$ |
| KD+CutMixPick (S. ent.) | $2.905_{\pm0.000}$ | $1.6556_{\pm0.0061}$ | $1.7509_{\pm0.0080}$ | $3.086_{\pm0.001}$ | $1.8355_{\pm0.0144}$ | $1.5419_{\pm0.0376}$ | $3.197_{\pm0.001}$ | $1.7290_{\pm0.0077}$ |
| KD+CutMixPick (T. ent.) | $2.681_{\pm0.005}$ | $1.6237_{\pm0.0094}$ | $1.6839_{\pm0.0159}$ | $2.812_{\pm0.002}$ | $1.8368_{\pm0.0060}$ | $1.4317_{\pm0.0096}$ | $2.948_{\pm0.004}$ | $1.6724_{\pm0.0113}$ |

the coefficients turn smaller while p-values larger than the cases on CIFAR100 and Tiny ImageNet. On ImageNet100, the p-values are below (or very close to) 5%, suggesting the correlation is still strong. On ImageNet, however, the results show a *weak* correlation between T. stddev and S. test loss, against our theory. Currently, we are not very sure why this happens sorely on ImageNet. After all, ImageNet100 is a subset of ImageNet and the proposed theory works fairly well on ImageNet100. One possible reason may be – full ImageNet historically was shown especially hard to make KD work, *e.g.*, in [7], they mentioned "*(Page 4) ...it is still a mystery why no teacher improves accuracy on ImageNet. Despite multiple recent papers in knowledge distillation, experiments on ImageNet are rarely reported. The few that do report find that standard setting of knowledge distillation fails on ImageNet [26] or perform an experiment with a small portion of ImageNet [21]*". Besides, we may notice the authors of CRD [45] mentioned in their GitHub issue[‡]: "*I have been struggling a bit to get KD work as well on ImageNet with ResNet-18 as the student network*". We conceive that the weak correlation on ImageNet may be related to this KD underperformance on ImageNet and may be related to the number of classes of the dataset. We shall continue to investigate this in our future work.

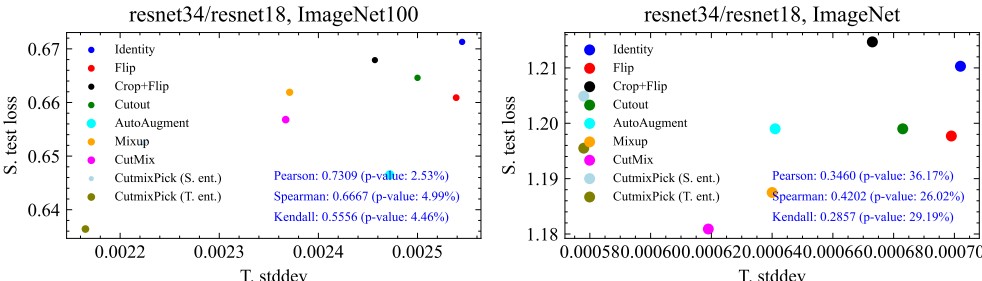

Figure 5: Scatter plots of T. stddev *vs.* S. test loss of different pairs on **ImageNet100** and **ImageNet**.

**Discussion: This paper discovers that "more correlation in the data, worse generalization ability". Is this an already-known fact in statistical learning [22, 47]?** Our work is *starkly different* from the established theory in [22, 47] in that they study the dependency *in the input data*,

[‡]https://github.com/HobbitLong/RepDistiller/issues/10#issuecomment-563078837

Table 7: Student test loss (S. test loss) and the stddev of teacher's mean probability (T. stddev, $\times 10^{-3}$) comparison on **ImageNet100** (a subset with 100 classes randomly drawn from ImageNet) when using different data augmentation schemes.

| Teacher | resnet34 | resnet34 |
|---|---|---|
| Student | / | resnet18 |
| Metric | T. stddev | S. test loss |
| KD+Identity | $2.545_{\pm 0.000}$ | $0.6713_{\pm 0.0029}$ |
| KD+Flip | $2.539_{\pm 0.004}$ | $0.6609_{\pm 0.0035}$ |
| KD+Crop+Flip | $2.457_{\pm 0.008}$ | $0.6679_{\pm 0.0027}$ |
| KD+Cutout | $2.500_{\pm 0.005}$ | $0.6646_{\pm 0.0030}$ |
| KD+AutoAugment | $2.472_{\pm 0.007}$ | $0.6465_{\pm 0.0078}$ |
| KD+Mixup | $2.371_{\pm 0.005}$ | $0.6619_{\pm 0.0043}$ |
| KD+CutMix | $2.367_{\pm 0.003}$ | $0.6568_{\pm 0.0048}$ |
| KD+CutMixPick (S. ent.) | $2.224_{\pm 0.006}$ | $0.6524_{\pm 0.0015}$ |
| KD+CutMixPick (T. ent.) | $2.165_{\pm 0.000}$ | $0.6364_{\pm 0.0049}$ |

Table 8: Student test loss (S. test loss) and the stddev of teacher's mean probability (T. stddev, $\times 10^{-3}$) comparison on **ImageNet** when using different data augmentation schemes. *The S. test losses are averaged by the last 5 epochs to mitigate the random variation.*

| Teacher | resnet34 | resnet34 |
|---|---|---|
| Student | / | resnet18 |
| Metric | T. stddev | S. test loss |
| KD+Identity | $0.702_{\pm 0.000}$ | 1.2103 |
| KD+Flip | $0.699_{\pm 0.000}$ | 1.1977 |
| KD+Crop+Flip | $0.673_{\pm 0.000}$ | 1.2147 |
| KD+Cutout | $0.683_{\pm 0.000}$ | 1.1990 |
| KD+AutoAugment | $0.641_{\pm 0.000}$ | 1.1990 |
| KD+Mixup | $0.640_{\pm 0.001}$ | 1.1875 |
| KD+CutMix | $0.619_{\pm 0.001}$ | 1.1809 |
| KD+CutMixPick (S. ent.) | $0.578_{\pm 0.000}$ | 1.2049 |
| KD+CutMixPick (T. ent.) | $0.578_{\pm 0.000}$ | 1.1955 |

while an important point in our theory is that we consider the *teacher's output* of the input, *not* the input per se. Namely, the proposed theory must be discussed *in the context of KD*. Existing works [22, 47] clearly are not in this scope.

**Use test accuracy *vs.* test loss as the measure of student's performance**. In the paper, we mentioned we use test loss instead of test accuracy as the measure of student's performance. The primary consideration, as discussed in the main paper, is that accuracy may be misaligned with loss sometimes (*i.e.*, ideally we expect *higher accuracy* coincides with *lower loss*; while in practice, we observe several counterexamples). Here we show an example of calculating the correlation between T. stddev and S. test *accuracy* in Fig. 6. As seen, if accuracy used, we observe no positive correlation between T. stddev and S. performance (which is the S. test accuracy here), while the correlation is strong if we use test loss. This shows the importance of using the *correct* measure for student's performance in a theoretical investigation of KD problems, as also noted by [29].

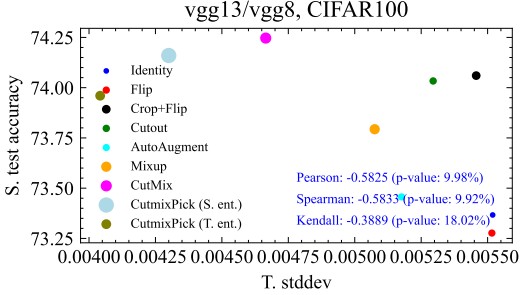

Figure 6: Showcase of using test *accuracy* instead of test *loss* as the measure of student's performance to conduct the T. stddev *vs.* S. performance correlation analysis in our paper.

**KD *vs*. CE loss on wrn_40_2/wrn_16_2 and vgg13/vgg8**. See Fig. 7. These plots show more examples that we can harvest considerable performance gain simply by using a strong DA with prolonged training.

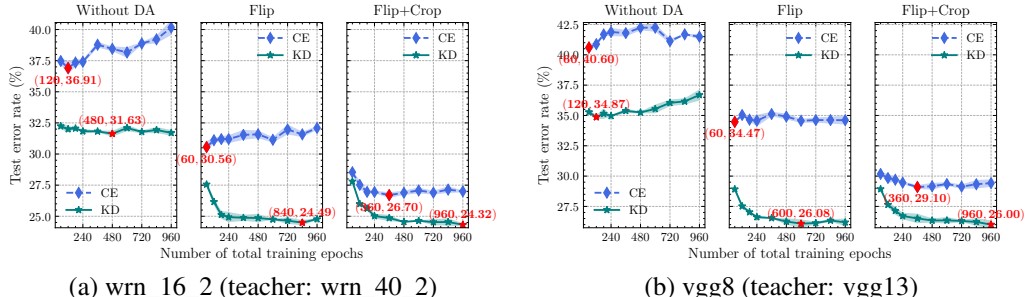

(a) wrn_16_2 (teacher: wrn_40_2)  (b) vgg8 (teacher: vgg13)

Figure 7: Test error rate of wrn_16_2 and vgg8 on CIFAR100 when trained for different numbers of epochs, using KD or cross-entropy (CE) loss, with or without data augmentation (DA). Every error rate is averaged by 3 random runs (shaded area indicates the stddevdev). Consistent with Fig. 2 in the main text, when DA is used, the optimal number of epochs is postponed and postponed more for KD than CE. When a stronger DA is used, the optimal number of epochs is postponed even more with smaller optimal test loss.

## A.2 More Explanations from Eq. (7) to Eq. (8)

In Proposition 3.1, the main idea is to have lower covariance among samples (in terms of the teacher's outputs), as suggested by Eq. (7). Then, it seems that the most straightforward idea is to use the covariance (or possibly the normalized covariance, *i.e.*, correlation) of the teacher's output as metric to capture the dependency among samples. Why does the paper not use this idea?

We first show this idea does not work as well as our proposed metric. Then we explain why.

Given a batch of input images, the teacher's output probability is $P \in \mathbb{R}^{N \times C}$, where $N$ is the batch size, $C$ is the number of classes. Then we consider the covariance and correlation of $P$ and try to construct a new metric from them:

$$
\begin{aligned}
\text{Cov}(P, P) \in \mathbb{R}^{N \times N} &\Rightarrow v = \text{Cov}(P, P).\text{mean}() \in \mathbb{R}, \\
\text{Cor}(P, P) \in \mathbb{R}^{N \times N} &\Rightarrow r = \text{Cor}(P, P).\text{mean}() \in \mathbb{R}.
\end{aligned}
\tag{12}
$$

We collect this (mean) covariance $v$ (a scalar) and correlation $r$ (a scalar) over many batches and then take the average of all the collected $v$ and $r$,

$$
\begin{aligned}
\bar{v} &= \frac{1}{K} \sum_{k \in [K]} v_k, \\
\bar{r} &= \frac{1}{K} \sum_{k \in [K]} r_k.
\end{aligned}
\tag{13}
$$

where $K$ is the total number of batches (10 epochs in our experiments, $K = 7,818$).

Then, we replace the proposed "T. stddev" with $\bar{v}$ and $\bar{r}$ in Tab. 3, resulting in two new tables: Tab. 9 and Tab. 10.

As seen, the foremost impression of Tab. 9 and Tab. 10 is that they have many red uparrows, which implies the metric goes *against* the supposed order, undermining its efficacy. *E.g.*, AutoAugment is stronger than Cutout (in that Autoaugment includes Cutout as a part; also AutoAugment performs better than Cutout as the test losses indicate). However, by $\bar{v}$ (see Tab. 9), AutoAugment is ranked "*weaker*" than Cutout with teacher resnet56 and vgg13.

As for Tab. 10, the correlation metric $\bar{r}$ is even worse than $\bar{v}$ in the sense that it incurs even more red uparrows than $\bar{v}$. Especially, CutMixPick (T. ent.) is the best DA (delivering the lowest test loss) while by $\bar{r}$ it "*underperforms*" all the other DA schemes except Identity and Autoaugment, with the teacher resnet56.

Table 9: Student test loss (S. test loss) and the **covariance of the teacher's output** ($\bar{v}$) comparison on CIFAR100 when using different DA schemes. This table is a replica of Tab. 3 just replacing the "T. stddev" with $\bar{v}$ here. The up (or down) arrow indicates the result is higher (or lower) than the one *right above it*. If "S. test loss" change is at a different direction from the "$\bar{v}$" change, we highlight the arrow in red.

| Teacher | wrn_40_2 | wrn_40_2 | wrn_40_2 | resnet56 | resnet56 | resnet56 | vgg13 | vgg13 | vgg13 |
| Student | / | wrn_16_2 | vgg8 | / | resnet20 | ShuffleV2 | / | vgg8 | MobileNetV2 |
| Metric | $\bar{v}$ | S. test loss | S. test loss | $\bar{v}$ | S. test loss | S. test loss | $\bar{v}$ | S. test loss | S. test loss |
|---|---|---|---|---|---|---|---|---|---|
| KD+Identity | 0.00015322 | 1.0976 | 1.1830 | 0.00014041 | 1.1783 | 0.9785 | 0.00015447 | 1.1856 | 1.6021 |
| KD+Flip | 0.00015258↓ | 1.0774↓ | 1.1673↓ | 0.00013956↓ | 1.1668↓ | 0.9961↑ | 0.00015522↑ | 1.1754↓ | 1.5779↓ |
| KD+Flip+Crop | 0.00014922↓ | 1.0837↑ | 1.1446↓ | 0.00013380↓ | 1.1763↑ | 0.9736↓ | 0.00015133↓ | 1.1537↓ | 1.5244↓ |
| KD+Cutout | 0.00014114↓ | 1.0564↓ | 1.1306↓ | 0.00012680↓ | 1.1687↓ | 0.9541↓ | 0.00014300↓ | 1.1279↓ | 1.4963↓ |
| KD+AutoAugment | 0.00013504↓ | 1.0305↓ | 1.1102↓ | 0.00013046↑ | 1.1478↓ | 0.9355↓ | 0.00014306↑ | 1.1273↓ | 1.4137↓ |
| KD+Mixup | 0.00012844↓ | 1.0507↑ | 1.0917↓ | 0.00011760↓ | 1.1621↑ | 0.9703↑ | 0.00013265↓ | 1.1194↓ | 1.4009↓ |
| KD+CutMix | 0.00011725↓ | 1.0310↓ | 1.0657↓ | 0.00010520↓ | 1.1424↓ | 0.9348↓ | 0.00011377↓ | 1.0728↓ | 1.2846↓ |
| KD+CutMixPick (S. ent.) | 0.00010184↓ | 1.0054↓ | 1.0471↓ | 0.00008972↓ | 1.1269↓ | 0.9339↓ | 0.00009727↓ | 1.0605↓ | 1.2739↓ |
| KD+CutMixPick (T. ent.) | 0.00009197↓ | 0.9753↓ | 0.9928↓ | 0.00007949↓ | 1.0853↓ | 0.9038↓ | 0.00008676↓ | 0.9884↓ | 1.1876↓ |

Table 10: Student test loss (S. test loss) and the **correlation coefficient of the teacher's output** ($\bar{r}$) comparison on CIFAR100 when using different DA schemes. This table is a replica of Tab. 3 just replacing the "T. stddev" with $\bar{r}$ here. The up (or down) arrow indicates the result is higher (or lower) than the one *right above it*. If "S. test loss" change is at a different direction from the "$\bar{r}$" change, we highlight the arrow in red.

| Teacher | wrn_40_2 | wrn_40_2 | wrn_40_2 | resnet56 | resnet56 | resnet56 | vgg13 | vgg13 | vgg13 |
| Student | / | wrn_16_2 | vgg8 | / | resnet20 | ShuffleV2 | / | vgg8 | MobileNetV2 |
| Metric | $\bar{r}$ | S. test loss | S. test loss | $\bar{r}$ | S. test loss | S. test loss | $\bar{r}$ | S. test loss | S. test loss |
|---|---|---|---|---|---|---|---|---|---|
| KD+Identity | 0.01570871 | 1.0976 | 1.1830 | 0.01570028 | 1.1783 | 0.9785 | 0.01562722 | 1.1856 | 1.6021 |
| KD+Flip | 0.01564524↓ | 1.0774↓ | 1.1673↓ | 0.01559539↓ | 1.1668↓ | 0.9961↑ | 0.01570224↑ | 1.1754↓ | 1.5779↓ |
| KD+Flip+Crop | 0.01559558↓ | 1.0837↑ | 1.1446↓ | 0.01559762↓ | 1.1763↑ | 0.9736↓ | 0.01559079↓ | 1.1537↓ | 1.5244↓ |
| KD+Cutout | 0.01530729↓ | 1.0564↓ | 1.1306↓ | 0.01510221↓ | 1.1687↓ | 0.9541↓ | 0.01531730↓ | 1.1279↓ | 1.4963↓ |
| KD+AutoAugment | 0.01856617↑ | 1.0305↓ | 1.1102↓ | 0.01971744↑ | 1.1478↓ | 0.9355↓ | 0.01947365↑ | 1.1273↓ | 1.4137↓ |
| KD+Mixup | 0.01497638↓ | 1.0507↑ | 1.0917↓ | 0.01512188↓ | 1.1621↑ | 0.9703↑ | 0.01507644↓ | 1.1194↓ | 1.4009↓ |
| KD+CutMix | 0.01498818↑ | 1.0310↓ | 1.0657↓ | 0.01502888↓ | 1.1424↓ | 0.9348↓ | 0.01473911↓ | 1.0728↓ | 1.2846↓ |
| KD+CutMixPick (S. ent.) | 0.01508117↑ | 1.0054↓ | 1.0471↓ | 0.01536088↑ | 1.1269↓ | 0.9339↓ | 0.01462629↓ | 1.0605↓ | 1.2739↓ |
| KD+CutMixPick (T. ent.) | 0.01525488↑ | 0.9753↓ | 0.9928↓ | 0.01564995↑ | 1.0853↓ | 0.9038↓ | 0.01487075↑ | 0.9884↓ | 1.1876↓ |

*Why does the covariance metric $\bar{v}$ not work as well as our proposed metric?* The covariance metric arises directly from our proposition, if the proposition is correct, why does it not work well? Fundamentally, the reason is that the covariance among samples is *hard* to estimate accurately. When we calculate the covariance matrix $\mathrm{Cov}(P, P)$, $P$ is of shape $N \times C$. Note, each row of $P$ is an *attribute (or random variable, RV)* and each column of $P$ is an *observation*[§]. The number of observations of $P$ is the number of classes, *which is a constant* for a given dataset (in the current case, it is 100 for the CIFAR100 dataset). Intuitively, given two random variables, in order to estimate the covariance between them, merely 100 observations is not very abundant. Namely, the limited observations render the direct estimation of the covariance among samples *inaccurate*. This can explain the counterexamples between $\bar{v}$ and test loss in Tab. 9.

(If the covariance metric is accurately estimated, it would not be surprising that the correlation in Tab. 10 is inaccurate, either. Especially, the normalization itself may not be desired. We put the results of using correlation as metric simply for a reference.)

*Then, how does the proposed metric resolve this "limited observation" problem?* The "smart thing" of the proposed metric is that it considers *the sum of RVs* instead of the RV itself. In statistics, it is well-known that, when multiple RVs are added together, the variance (or equivalently, stddev) of the sum RV will take into account the covariance among its members, as shown below:

$$\mathrm{Var}(\sum_{i=1}^{N} X_i) = \sum_{i=1}^{N} \mathrm{Var}(X_i) + 2 \sum_{1 \leq i < j \leq N} \mathrm{Cov}(X_i, X_j), \tag{14}$$

where $X_i$ (with a little abuse of notation here) is the teacher's output probability of the $i$-th example.

---

[§]Interested readers are encouraged to check out this numpy covariance function, which we use to implement $\bar{v}$: https://numpy.org/doc/stable/reference/generated/numpy.cov.html

Essentially, we want to estimate the covariance term in RHS (*i.e.*, $\sum_{1 \leq i < j \leq N} \text{Cov}(X_i, X_j)$). Now, what we do in the paper is to use the LHS (*i.e.*, $\text{Var}(\sum_{i=1}^{N} X_i)$) as a *proxy* of the covariance term. Clearly, there is a *hidden assumption* here to allow us to use such a proxy: the first term of RHS (*i.e.*, $\sum_{i=1}^{N} \text{Var}(X_i)$) stays (nearly) the same for different samples (*i.e.*, different batches) of $\{X_i\}$. In our work, there are two conditions to make this hidden assumption hold in practice. **(1)** The augmented images are based on the original images. We can still consider them in the same domain (in our Proposition 3.1, $X_i$ is actually assumed to be drawn from the same distribution), so $\text{Var}(X_i)$ is close for different $X_i$. **(2)** We have abundant samples ($N = 640$ in our experiments for CIFAR100 and Tiny ImageNet). The sum effect of so many samples makes $\sum_{i=1}^{N} \text{Var}(X_i)$ stabilize to a constant.

We can actually come up with counterexamples that *intentionally* break the hidden assumption. *E.g.*, consider a "bad" DA which turns all input images to a *constant* image. Then the input batch will become a repetition of just one image. When we input such a batch to the teacher, the teacher's output probability will be the same. Then the $\text{Var}(\sum_{i=1}^{N} X_i)$ term will become zero. It cannot be a legitimate proxy for the covariance term anymore as the first term of RHS now is zero.

We are aware of the existence of such counterexamples. Notably, they do *not* affect the validity of the proposed metric. Here are the reasons – For one thing, such bad DA rarely appears in practice. The strong correlation per se already demonstrates the efficacy of the proposed metric in practical cases. For another thing, there is an important design in our training setup to avoid such counterexamples – Note we *add* the augmented images to the original images instead of replacing them (see Fig. 3). Therefore, in a batch, it is *unlikely* that all the images degenerate to one image. Namely, the counterexample we just mentioned never appear in practice.

Then how to understand such counterexample? Why it *appears* to go against our theory? Fundamentally, the counterexample we just mentioned breaks one of the key assumptions of our proposition in the first place – each individual sample (including the original image *and the augmented image*) obeys the same true (unknown) data distribution $\mathcal{D}$. The bad DA transforms all input images to a constant image, which already means the individual sample is not drawn from the true data distribution since randomly drawing from the true data distribution will not give us the same image.

To further confirm this, we implement the "bad" DA and re-check the validity of the proposed metric $\bar{m}$ with wrn_40_2/wrn_16_2 on CIFAR100. When using the "bad" DA, we have the metric value $\bar{m} = 0.022439$. Compare this to Tab. 3, we will notice it is ranked the *highest* (*i.e.*, the *worst*). This agrees with our expectation.

In short, the counterexamples above do not affect the efficacy of our proposed metric in practice.

### A.3 Example of Calculating T. Stddev

Take CIFAR100 for an example. It has 50,000 training samples. With batch size 64, one epoch amounts to 782 batches. During training, the $K$ in Eq. (9) is set to 10. During the iteration of training data loader, we collect the teacher's outputs. Every $K/2$ training batches (the division of 2 is because we *append* the augmented images to the original images), we obtain a matrix of shape $[K \times \text{batch\_size}, \text{num\_classes}]$. Then we average this matrix along the 1st axis to give us a vector, *i.e.*, the $\mathbf{u}$ in Eq. (9). In brief, every $K/2$ batches, we have one sample of $\mathbf{u}$. In total, we run the data loader for *10 epochs* (10 is empirically set; more epochs should give us more accurate estimation but we just find 10 is good enough), which gives us 7820/5=1654 $\mathbf{u}$'s, *i.e.*, a matrix of shape $[1654, \text{num\_classes}]$. Then we calculate the variance along the 1st axis, giving us a vector of shape $[\text{num\_classes}]$, *i.e.*, the $\mathbf{m}$ in Eq. (10). Finally, we average the stddev elements in $\mathbf{m}$ to obtain the proposed *scalar* metric.

For more details, please check our code at: http://github.com/MingSun-Tse/Good-DA-in-KD.

### A.4 CutMix Sample Analysis on ImageNet

**CutMix sample analysis and why KD is naturally suited to exploit CutMix**. During the KD training of resnet34/resnet18 on ImageNet, we recorded the CutMix samples on which the teacher *disagrees* with the CutMix scheme on the label. We call this *label disagreement issue*.

As show in Fig. 8. there exist cases where the image cut from one image covers the salient object in the other. For example, the cab in (a) completely covers the ground beetle. In this case, using

the label by CutMix does not make sense anymore. A similar problem appears on (b). Note that these misleading labels by CutMix are rectified when the teacher is employed to guide the student. The teacher assigns the correct label "cab" to (a) and "Yorkshire terrier" to (b) (which is still not the true label "Tibetan terrier" but it is clearly more relevant and "Tibetan terrier" is also in the top-5 predictions). For (c) and (d), they pose a problem more than occlusion: the foreground cut in (c) is labeled as "acoustic guitar", however, the cut is too small for us to make it out without knowing the label. Meanwhile, the background object "Arabian camel" is occluded. Then the grids in the picture turn out to be the most salient part. If we look at the predictions of the teacher, "shopping cart" and "shopping basket" clearly make more sense than either of the original two labels. A similar issue happens on (d), where the "Indian elephant" is largely occluded. The foreground cut is labeled "quill" but the bottle in the middle is more salient. Thus the teacher predicted it as "coffeepot", "milk can", *etc*.

In order to see how severe the label disagreement issue is, we counted the number of these synthetic samples and found that on **more than half of the samples (52.1%)** produced by CutMix, the teacher model and CutMix hold a different view regarding the label. Many of these suffer from the problem shown in Fig. 8. The KD loss can rectify these label mistakes. This further shows the interplay between KD and DA: KD thrives on DA and *in turn, some DA schemes are more reasonable for KD* (than CE) where a teacher can supply more relevant labels.

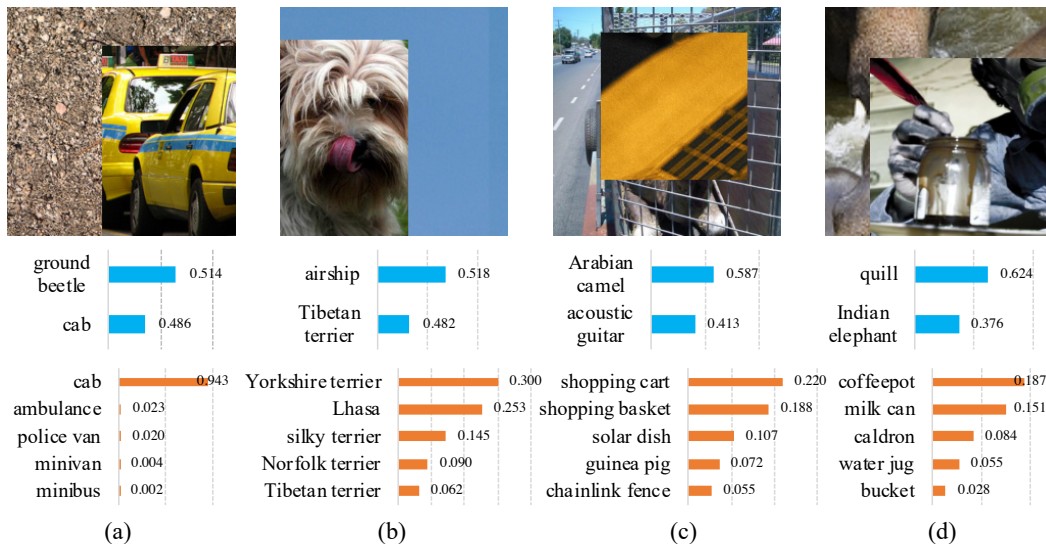

Figure 8: ImageNet CutMix samples where the main object in one of the images is no longer visible after CutMix augmentation. Below each sample, the first is the target probability assigned by CutMix and the second is the top-5 predicted probabilities by the teacher. These examples can be misleading when cross-entropy loss is used, but not for KD, as explained in the text.

## A.5   Dataset License and Hardware Condition

The four datasets used in this paper (CIFAR100, Tiny ImageNet, ImageNet100, ImageNet) are all publicly available. One KD experiment with one NIVIDA 2080Ti GPU on CIFAR100 takes around 2 to 6 hrs; while an experiment on Tiny ImageNet, ImageNet100, or ImageNet may take up to 24 hrs on the 2080Ti GPU. In general, *the experimenting is not compute intensive*. For details, please refer to our GitHub code: http://github.com/MingSun-Tse/Good-DA-in-KD.

## A.6   Limitations and Potential Negative Societal Impacts

**Limitations**. The proposed metric is calculated with finite training samples. The precision of the metric values will necessarily depend on the number of training samples used. In this regard, we have tried our best to avoid the influence of statistical variations (using abundant samples). The presented results in the paper are shown stable. Especially, the correlation in Tabs. 1 and 2 is strong. This fact itself is a proof that the limitation of our paper in this regard is small.

**Potential Negative Societal Impacts**. Simply put, this work proposes theories and algorithms that can boost the performance of a network with the aid of a larger network (*i.e.*, the teacher-student distillation), that is, make it smaller, faster, and possibly consume less energy in practical applications. We focus on the classification task, which is generally the foundation of the many up-stream computer vision tasks like detection and segmentation. Therefore, this work potentially has a broad application especially in the computer vision areas.

The algorithm itself has few negative societal issues, but when it makes many AI-driven technologies applicable in practice, the impact really depends on how humans use them. This actually falls into the general ethical discussion on whether AI is good or not. Beyond this scope, this work does not have specific negative societal impacts brought by its potential application, to our best knowledge.