# OpenReview forum: "What Makes a "Good" Data Augmentation in Knowledge Distillation - A Statistical Perspective"
_NeurIPS.cc/2022/Conference — NeurIPS 2022 Accept_

### Official Review · Reviewer_VLkK · 2022-07-05

**Rating:** 5
**Confidence:** 3
**Soundness:** 2 fair
**Presentation:** 2 fair
**Contribution:** 2 fair

**Summary:**

The major contributions of this paper are 2-fold:
1) The authors suggest that given a fixed teacher model, a good data augmentation (DA) scheme is characterized by a lower variance of the teacher’s mean output probability. This claim is supported by statistical analysis and empirical results (image classification).

2) An entropy-based sample picking scheme is introduced to synthetically reduce the variance of the teacher’s mean probability. This scheme (CutMixPick) further improves the results.

**Questions:**

Please see Weaknesses section above for a list of all questions.

Remark : I managed to only study the main text. Unfortunately, I only skimmed through the supplementary, but didn't study in detail due to its length / my time-constraints.

**Limitations:**

The authors have discussed limitations / potential societal impacts in Supplementary Section 6.

**Strengths And Weaknesses:**

**Strengths:**
1) Setups with lower variance (although it is not clear how $u$ is calculated, see weaknesses below) obtains acceptable improvements in CIFAR-100 and Tiny-ImageNet experiments.


**Weaknesses:**

Though this work is executed well and the image classification results show improvements, the novelty and significance of this work are limited. See below:
1) The clarity of Section 3.2 can be improved. Though proposition 1 seems interesting, I have questions regarding the Std of the teacher’s mean output probability. It is unclear as to how $u$ in equation 9 is calculated. Particularly:
- What is $S^*$ and how is it determined? Why is $K=640$?
- Would it be reasonable to calculate $m$ using the variance of the output probabilities of the teacher **for the correct class** over all samples?
2) Experiment coverage / results are limited:
- Particularly, ImageNet results are not discussed and analyzed. Not that I have anything against CIFAR-100, Tiny-ImageNet analysis, but I tend to think that ImageNet results are critical to add merit to the findings (1000 classes, higher resolution:224 x 224).  Most KD works report results on ImageNet-1K ([1, 2, 3, 4, 5]). If compute is a problem, authors can consider using a smaller dataset for experiments: ImageNet-100 (A subset of ImageNet popular in the self-supervised learning community). I believe the authors can also use publicly available pre-trained ImageNet models if required: https://pytorch.org/vision/stable/models.html
- What is the reason for not including Flip, Flip+Crop, Cutout, Auto-Augment, Mixup results in Tables 3, 4?
3) For ‘Std of mean prob’ in Tables 1, 2, is it reasonable to use the training set as the training set is used for distillation (not the test set)?
4) For CutMixPick experiments, what is the percentage of the training set used for distillation (after 'picking' / filtering)?

Overall I enjoyed reading this paper. In my opinion, the weaknesses of this paper outweigh the strengths. But I’m willing to change my opinion based on the rebuttal.


=====================

[1] Müller, Rafael, Simon Kornblith, and Geoffrey E. Hinton. "When does label smoothing help?." Advances in neural information processing systems 32 (2019).

[2] Shen, Z., Liu, Z., Xu, D., Chen, Z., Cheng, K. T., & Savvides, M. (2021). Is label smoothing truly incompatible with knowledge distillation: An empirical study. In ICLR

[3] Chandrasegaran, K., Tran, N. T., Zhao, Y., & Cheung, N. M. (2022). Revisiting Label Smoothing and Knowledge Distillation Compatibility: What was Missing?. ICML

[4] Heo, Byeongho and Kim, Jeesoo and Yun, Sangdoo and Park, Hyojin and Kwak, Nojun and Choi, Jin Young (2019). A Comprehensive Overhaul of Feature Distillation. ICCV

[5] Tang, Jiaxi, et al. "Understanding and improving knowledge distillation." arXiv preprint arXiv:2002.03532 (2020).

---

> ### Author Response · Authors · 2022-08-02
> **Responses to Reviewer VLkK (Part 2)**
>
> `Q3`: For ‘Std of mean prob’ in Tables 1, 2, is it reasonable to use the training set as the training set is used for distillation (not the test set)?
>
> `A3`: We think it is reasonable because the metric is used to select the best DA when we attempt to enhance some KD algorithms. Training set is the set for training/develping algorithm. If we use the test set, it may be at the risk of peeking at test data and unfair.
>
> `Q4`: For CutMixPick experiments, what is the percentage of the training set used for distillation (after 'picking' / filtering)?
>
> `A4`: There might be a little misunderstanding here. We *append* the augmented images to the original training set (see Line 182-183), so *all* of the training set is used for distillation. We assume what you actually mean is: in each training batch, what percentage is for the original training images? The answer is 50%, see Line 206 - 207.
>
> **Lastly, thank you so much for helping us improve the paper so far! Please let us know *asap* if you have any further questions. We are actively available during this rebuttal!**

---

> ### Author Response · Authors · 2022-08-02
> **Responses to Reviewer VLkK (Part 1)**
>
> Thank you so much for your constructive comments! We address your concerns as follows.
>
> `Q1.1`: What is $S^*$ and how is it determined? Why is $K=640$?
>
> `A1.1`: $S^*$ is the sampled example set (i.e., a bunch of training examples) to estimate $m$ in Eq. 10. During training, we cache several batches (10 batches) of training data to make $S^*$. $K=640$ because we sample 10 batches during training. For the CIFAR100 and Tiny ImageNet datasets, batch size is 64, thus 64*10=640.
>
> Thank you so much for pointing these unclear parts out! We’ll clarify them in our revised version.
>
> `Q1.2`: Would it be reasonable to calculate $m$ using the variance of the output probabilities of the teacher for the correct class over all samples?
>
> `A1.2`: $m$ is to estimate the variance of $u$, which is to estimate the $p^{(t)}(x)$ in Eq. 9 (or its prior form in Eq. 8). You can tell from Eq.9 or Eq. 8: the random variable is *only about the teacher and input, no need for the labels*. Thus, rigorously, there is no such "*correct class*" at all. The metric is proposed according to the theory. Since the theory does not imply using “correct class” to calculate $m$, then we believe it is more reasonable not to do so.
>
> `Q2.1`: Particularly, ImageNet results are not discussed and analyzed. Not that I have anything against CIFAR-100, Tiny-ImageNet analysis, but I tend to think that ImageNet results are critical to add merit to the findings (1000 classes, higher resolution:224 x 224). Most KD works report results on ImageNet-1K ([1, 2, 3, 4, 5]). If compute is a problem, authors can consider using a smaller dataset for experiments: ImageNet-100 (A subset of ImageNet popular in the self-supervised learning community). I believe the authors can also use publicly available pre-trained ImageNet models if required: https://pytorch.org/vision/stable/models.html
>
> `A2.1`: Thank you so much for letting us know about these relevant papers! *We will cite and discuss them in our related work*.
>
> Meanwhile, we report new results here following your suggestion: Use ImageNet-100 (a subset of 100 classes randomly drawn from the original ImageNet. The image size is 224\*224\*3). The teacher is ResNet34 and student is ResNet18. Then we reproduce Tab. 1 (or 2) with the **ResNet34/ResNet18 pair on ImageNet100**. Results are below. The student test error is *averaged by 3 random runs*.
>
> |  | Std of mean prob. | Test err. |
> |---|---|---|
> | KD+Identity | 0.002545 | 0.6713$_{\pm0.0029}$ |
> | KD+Flip | 0.002543 | 0.6609$_{\pm0.0035}$ |
> | KD+Flip+Crop | 0.002456 | 0.6679$_{\pm0.0027}$ |
> | KD+Cutout | 0.002505 | 0.6646$_{\pm0.0030}$ |
> | KD+AutoAugment | 0.002463 | 0.6465$_{\pm0.0078}$ |
> | KD+Mixup | 0.002365 | 0.6619$_{\pm0.0043}$ |
> | KD+CutMix | 0.002363 | 0.6573$_{\pm0.0055}$ |
> | KD+CutMixPick (S ent.) | 0.002215 | 0.6524$_{\pm0.0015}$ |
> | KD+CutMixPick (T ent.) | 0.002166 | 0.6356$_{\pm0.0054}$ |
>
> As seen, there are a few cases that go against the proposed theory. E.g., Flip+Crop has a lower "Std of mean prob." than Flip, while the student error is higher. This means that large-size dataset like ImageNet-100 is indeed more challenging and making the validation of the theory harder. This said, the general trend is still aligned with our expectation: *The proposed method CutMixPick achieves the lowest "Std of mean prob.", also, the lowest test student error*, implying it can generalize to the large-size dataset.
>
> **[ 08/08 Edited ]** we calculate the correlation coefficients (p-values) of the above ImageNet-100 results, shown below. All the p-values are still *below 5%, which means the correlation is literally **statistically significant*** even on this challenging dataset.
>   - Pearson: 0.7331 (0.02463)
>   - Spearman:  0.6667 (0.04987)
>   - Kendall: 0.5556 (0.04462)
>
> `Q2.2`: What is the reason for not including Flip, Flip+Crop, Cutout, Auto-Augment, Mixup results in Tables 3, 4?
>
> `A2.2`: Since we have shown in Tab.1  and 2 that the best DA is CutMix, we only include the results of CutMix in Tab. 3 and 4 (to save computation) when we prepared the submission. During this rebuttal, we do not have enough time/resources to make up them, but we **promise to include them all in the revised version**. And note, even without these results, Tab. 1 and 2 (also the above new results on ImageNet-100) have well validated the proposed proposition and made the paper self-contained.

---

> > ### Comment · Reviewer_VLkK · 2022-08-08
> > **Reply**
> >
> > Thank you authors for the great effort on the rebuttal. Authors have addressed my concerns.
> >
> > As I pointed out (and very nicely shown by the authors in rebuttal), when applying the proposed theory to practical problems (i.e.: ImageNet-100), there are cases that go against the proposed theory. Therefore, I’m not certain regarding the applications of the proposed theory although I acknowledge that the proposed CutmixPick achieves the lowest Std of mean prob and lowest student test error.

---

> > > ### Author Response · Authors · 2022-08-08
> > > **Thank you for following up!**
> > >
> > > Thank you so much for the feedback!
> > >
> > > We understand your point! Meanwhile, we would like to mention that
> > > - First, it is normal in research that we see more violations when we extend the theory to a more complex practical application (that is exactly why we call it "more complex").
> > > - Second, historically, making KD effective on ImageNet is non-trivially hard (see [this screenshot from [*1]](https://tva1.sinaimg.cn/large/e6c9d24egy1h5014ag2ecj20g00amdia.jpg)). Note, despite this, our theory still helps us identify the best DA scheme CutMix-Pick on ImageNet-100 and its superiority aligns well with the proposed metric.
> > > - Third, we calculate the correlation coefficients (p-values) of the above ImageNet-100 results, shown below. All the p-values are still *below 5%, which means the correlation is literally **statistically significant*** even on this challenging dataset.
> > >   - Pearson: 0.7331 (0.02463)
> > >   - Spearman:  0.6667 (0.04987)
> > >   - Kendall: 0.5556 (0.04462)
> > >
> > > As we know, it is hard to get a theory that works for all cases. If our responses have resolved your major concerns, we sincerely hope you may pay more attention to the pros of our paper and generously consider raising the score even a little bit in recognition of our efforts so far. Thank you *so much* for helping us improve our paper!
> > >
> > > [*1] On the Efficacy of Knowledge Distillation, ICCV, 2019

---

> > > > ### Comment · Reviewer_VLkK · 2022-08-09
> > > > **Reply2**
> > > >
> > > > Thank you. I have reviewed the details in the image you posted, but this work [1] shows otherwise: KD can improve students when using ImageNet. In fact, this work uses only soft-targets from the teacher (no ground-truth labels) for KD which is **contradictory to the details you posted**. I.e.: ResNet-50 $\rightarrow$ ResNet-18 KD obtains 71.425% whereas ResNet-18 training from scratch obtains only 69.758%. ( See Tables 1, 2 in [1] ). See https://pytorch.org/vision/stable/models.html for official pytorch model weights / accuracy. Please check.
> > > >
> > > > In the revised version, please consider discussing ImageNet results.
> > > >
> > > > **Considering the efforts made by the authors, I have increased my recommendation, although I stand by my review and consider this work to be borderline.**
> > > >
> > > > [1] Shen, Z., Liu, Z., Xu, D., Chen, Z., Cheng, K. T., & Savvides, M. (2021). Is label smoothing truly incompatible with knowledge distillation: An empirical study. In ICLR

---

> > > > > ### Author Response · Authors · 2022-08-09
> > > > > **Thank you!**
> > > > >
> > > > > > Thank you. I have reviewed the details in the image you posted, but this work [1] shows otherwise: KD can improve students when using ImageNet. In fact, this work uses only soft-targets from the teacher (no ground-truth labels) for KD which is contradictory to the details you posted. I.e.: ResNet-50  ResNet-18 KD obtains 71.425% whereas ResNet-18 training from scratch obtains only 69.758%. ( See Tables 1, 2 in [1] ). See https://pytorch.org/vision/stable/models.html for official pytorch model weights / accuracy. Please check.
> > > > >
> > > > > > In the revised version, please consider discussing ImageNet results.
> > > > >
> > > > > Yes, exactly! Now we know KD can also work on ImageNet thanks to the very inspiring works like [1]. We mentioned the ICCV'19 paper simply to show it *was* non-trivial to make KD work on ImageNet back then and ImageNet is indeed *harder* for KD. (We'll cite [1] and indicate its remarkable KD performance on ImageNet in our paper).
> > > > >
> > > > > In general, we are *so grateful* that you improved the score, which is a great encouragement to us! As suggested, we will include the full ImageNet results in the revised version. **Again, thank you so much!**

---

### Official Review · Reviewer_Tm5Y · 2022-07-06

**Rating:** 7
**Confidence:** 4
**Soundness:** 4 excellent
**Presentation:** 4 excellent
**Contribution:** 4 excellent

**Summary:**

The paper presents a theoretical development of a straightforward to use measure of effectiveness of data augmentation techniques for distillation. It proposes that this measure need only be based on an analysis of the teacher, without consideration of the student. The paper also presents an enhanced version of the cut-mix data augmentation technique, which performs a distillation of the augmented dataset to maximize the information content of the data.

**Questions:**

Non-data manipulations, such as drop-out, can result in the same behavior as shown in figure 2a (of spreading different soft targets while keeping the same hard targets).
The paper (Bouthillier, X., Konda, K., Vincent, P., and Memisevic, R. Dropout as data augmentation. arXiv preprint arXiv:1506.08700, 2015.) even considers the dropout as a type of data augmentation. So, could your approach be made even more general (than the few types of augmentation you consider)?

In lines 127-129, why is a script L used to indicate the log function, rather than the more usual log() notation? Script letters are conventionally used to indicate spaces or sets. In line 139 the script L is used to indicate a loss function, adding to the confusion.

In line 154-156 the assumption is made of a "well-trained" student, that gives softmax outputs "pretty equal" to those of the teacher. However, in most practical applications of distillation, the student can be much smaller than the teacher (e.g. for network compression for edge devices). What happens to your conclusions in that case? This assumption is used to make the claim that we need only look at the teacher to judge the effectiveness of a DA method. So, is this claim still true if the assumption does not hold?
In table 1, (as was done in table 3) you should give the teacher and student test accuracies to see how well the "well-trained" assumption holds. For CIFAR-100 and the student/teacher networks used, I would expect a significant difference in softmax outputs (e.g. 74% for ResNet56 and 69% for ResNet20).

It seems surprising that an analysis of the student has no role to play in determining the effectiveness of a data augmentation technique that is being used to train that student. If no distillation was being done at all (just training using cross-entropy on the hard targets), DA still would be of some use, no? So, this suggests that one cannot just ignore the student in determining the effectiveness of the DA method. Perhaps, in light of the results in table 1, the teacher's effects dominate the student's effects in terms of judging DA performance, but the student may still have some influence. It would be nice to have some theoretical insight into why this is, rather than just throwing away the student based on the "well-trained" assumption.

Does the "pick" aspect of the Cut-Mix-Pick help with the other DA methods? Presumably doing the data distillation on the other augmentations would also improve their results.

In section 5.1 the authors say that they do not look at accuracy, as the derivations are based on error, and sometimes a network can give high accuracy but also high error. But accuracy is typically what people are interested in when using networks, not the value of the loss function. It may be that by focusing on loss (error) one is biasing the student to become more confident, like the teacher, and perhaps overly confident, leading to reduced accuracy. In any case, table 3 shows accuracies, so I am not sure why this was not done in section 5.1.

**Limitations:**

There is no discussion of limitations and potential negative societal impacts. However, distillation techniques, to the extent they can be used to train smaller networks, may positively affect societal impacts related to high power usage during training and inference time.

**Strengths And Weaknesses:**

The method yields an informative measure that, empirically, can be reliably used to judge the efficacy of different data augmentation schemes, at least for the methods considered.

The justification and derivation of the proposed measure is reasonable, but it (possibly) depends on an overly strong assumption on the student to be "well-trained". This may not be the case in some applications where the student has a much lower capacity than the teacher. However, the proposed measure may still be effective, but would need more detailed theoretical justification to show this.

The cut-mix-pick data augmentation technique demonstrates improved performance over the standard cut-mix method.

---

> ### Author Response · Authors · 2022-08-02
> **Responses to Reviewer Tm5Y (Part 2)**
>
> `Q4`: It seems surprising that an analysis of the student has no role to play in determining the effectiveness of a data augmentation technique that is being used to train that student. If no distillation was being done at all (just training using cross-entropy on the hard targets), DA still would be of some use, no? So, this suggests that one cannot just ignore the student in determining the effectiveness of the DA method. Perhaps, in light of the results in table 1, the teacher's effects dominate the student's effects in terms of judging DA performance, but the student may still have some influence. It would be nice to have some theoretical insight into why this is, rather than just throwing away the student based on the "well-trained" assumption.
>
> `A4`: We totally agree that the student plays a role when deciding which DA is better specific to that student. Ideally, each DA has to be evaluated by *a specific pair of teacher and student* to say whether it is better or worse than the other one -- this is exactly what Eq. 8 tells before the approximation. Yet clearly, in practice, when we try to put our theory into use, we cannot just evaluate all the pairs -- If we already run all the experiments and have the trained students, we just know which DA performs best from the student accuracy. It would be no meaning to calculate Eq. 8 (or other equations) anymore.
>
> We will revise the statement to clarify this confusion. Thanks for letting us know about this issue!
>
> `Q5`: Does the "pick" aspect of the Cut-Mix-Pick help with the other DA methods? Presumably doing the data distillation on the other augmentations would also improve their results.
>
> `A5`: In table below, we report the **student test errors** of applying picking to other DA methods in addition to CutMix on CIFAR100, VGG13/VGG8 pair. $\pm$ indicates the result is averaged by 3 random runs (mean and std reported). ***We promise to release all code/checkpoints/logs.***  **[Edited 08/03:]** The results have been updated as follows. Thanks!
>
> | DA method | W/O picking | W/ picking |
> |---|---|---|
> | Identity | 1.1856$_{\pm0.0196}$ | 1.1922$_{\pm0.0075}$ |
> | Flip | 1.1754$_{\pm0.0105}$ | 1.1843$_{\pm0.0075}$ |
> | Flip+Crop | 1.1537$_{\pm0.0059}$ |  1.1824$_{\pm0.0096}$ |
> | Cutout | 1.1279$_{\pm0.0055}$ | 1.0760$_{\pm0.0087}$ |
> | Autoaugment | 1.1273$_{\pm0.0213}$ |  1.0231$_{\pm0.0028}$ |
> | Mixup | 1.1194$_{\pm0.0176}$ | 1.0323$_{\pm0.0119}$ |
>
> The "W/O picking" results are from Tab. 1 in the paper (std is added). "W/ picking" is new. As seen, the proposed picking scheme (Sec. 4.2) also improves other popular data augmentations like Cutout, Autoaugment, and Mixup. But it does not improve upon Identity/Flip/Flip+Crop. This is not surprising because the default DA is Flip+Crop; the augmented samples are actually pretty similar to the original ones. Then, picking the samples with higher entropy will select out those similar samples, causing the covariance in a batch to increase, which will lead to higher student error per our theory .
>
> `Q6`: In section 5.1 the authors say that they do not look at accuracy, as the derivations are based on error, and sometimes a network can give high accuracy but also high error. But accuracy is typically what people are interested in when using networks, not the value of the loss function. It may be that by focusing on loss (error) one is biasing the student to become more confident, like the teacher, and perhaps overly confident, leading to reduced accuracy. In any case, table 3 shows accuracies, so I am not sure why this was not done in section 5.1.
>
> A6: We agree that accuracy is used more often in practice. Yet when we attempt to theoretically answer the question “what makes a good DA in KD”, it might be better to make the empirical validation *exactly the same* as how the theory is developed so that we can know whether the theory is correct or wrong.
>
> Lower loss typically implies higher accuracy, yet not necessarily. We observe this in our experiments and also noted by prior works (like [19]).
>
> The inconsistency between Tab. 3/4 and 1/2 causes confusion. Thank you for letting us know this! We will
> - change all the measures to test loss in Tab 3/4 to make them consistent.
> - report the accuracy results in our supplementary material (to confirm that accuracy may not be well aligned with loss)
> - discuss how we can extend the proposition to using accuracy as the measure for generalization
>
> **Lastly, thank you so much for helping us improve the paper so far! Please let us know *asap* if you have any further questions. We are actively available during this rebuttal!**

---

> ### Author Response · Authors · 2022-08-02
> **Responses to Reviewer Tm5Y (Part 1)**
>
> Thank you so much for your constructive comments! We address your concerns as follows.
>
> `Q1`: Non-data manipulations, such as drop-out, can result in the same behavior as shown in figure 2a (of spreading different soft targets while keeping the same hard targets). The paper (Bouthillier, X., Konda, K., Vincent, P., and Memisevic, R. Dropout as data augmentation. arXiv preprint arXiv:1506.08700, 2015.) even considers the dropout as a type of data augmentation. So, could your approach be made even more general (than the few types of augmentation you consider)?
>
> `A1`: It might be hard to include dropout into our framework now since dropout does not explicitly change the inputs. It would be interesting to make our approach more general, yet currently, the paper in its present form may not allow us to claim so broadly. So we prefer to limit the scope of data augmentation in this paper to the conventional ones (i.e., input transformation or input mixing) to avoid overclaiming.
>
> Thanks for pointing this out! We will revise the paper to make the DA concept more rigorous and cite the paper you mentioned to have a discussion regarding if we can extend the DA scope to broader.
>
> `Q2`: In lines 127-129, why is a script L used to indicate the log function, rather than the more usual log() notation? Script letters are conventionally used to indicate spaces or sets. In line 139 the script L is used to indicate a loss function, adding to the confusion.
>
> `A2`: Thank you so much for letting us know about this problem. We’ll use the $\log$ notation following your suggestion!
>
> `Q3`: In line 154-156 the assumption is made of a "well-trained" student, that gives softmax outputs "pretty equal" to those of the teacher. However, in most practical applications of distillation, the student can be much smaller than the teacher (e.g. for network compression for edge devices). What happens to your conclusions in that case? This assumption is used to make the claim that we need only look at the teacher to judge the effectiveness of a DA method. So, is this claim still true if the assumption does not hold? In table 1, (as was done in table 3) you should give the teacher and student test accuracies to see how well the "well-trained" assumption holds. For CIFAR-100 and the student/teacher networks used, I would expect a significant difference in softmax outputs (e.g. 74% for ResNet56 and 69% for ResNet20).
>
> `A3`: "*What happens to your conclusions in that case?*" -- If the student is small or not well-trained, the KL div term will become large (think about a naive case: the student is completely a random model). Yet the covariance captures the dependency among the training examples. Given any fixed function f(*), if two inputs x1 and x2 are highly correlated,  f(x1) and f(x2) will also be highly correlated, which will still make the DA a “bad” one.
>
> We lift the student term in Eq. 8 simply for a more neat/convenient form of the proposed metric when it is used in practice (no need for training the student for real). The assumption is brought up for practical use (for the metric). Even if the assumption (in Eq. 8) does not hold, our theory (Proposition 1) still holds; just the metric (Eq. 10) may not hold (for the “bad” student).
>
> When it comes to practice, we do have the freedom to choose which student is used in Eq. 8. But clearly, the top-performing students are more favored. E.g., given a random student, it ranks DA1 better than DA2; meanwhile, given a well-trained student, it ranks DA2 better than DA1. Obviously, we will choose DA2 as it relates to a better student.
>
> “*In table 1, (as was done in table 3) you should give the teacher and student test accuracies to see how well the "well-trained" assumption holds*” -- The accuracies in Tab. 1 are reported below:
> - WRN-40-2: 75.61, WRN-16-2: ~75.3,  VGG8: ~74.3
> - ResNet56: 72.34, ResNet20: ~70.7, ShuffleNetV2: ~75.9
> - VGG13: 74.64, VGG8: ~74.1, MobileNetV2: ~68.5
>
> where the teacher’s accuracies are exact. The student accuracies are averaged across different data augmentation schemes. As seen, there is the case where teacher and student have a gap (like VGG13/MobileNetV2). Nevertheless, they do not affect the empirical validation of our proposition.

---

> > ### Comment · Reviewer_Tm5Y · 2022-08-07
> > **"well trained student"**
> >
> > Thank you for your answers. I am still concerned about the assumption of a "well-trained student". Your answer
> > "We lift the student term in Eq. 8 simply for a more neat/convenient form of the proposed metric when it is used in practice (no need for training the student for real). The assumption is brought up for practical use (for the metric). Even if the assumption (in Eq. 8) does not hold, our theory (Proposition 1) still holds; just the metric (Eq. 10) may not hold (for the “bad” student)."
> >
> > Even though Proposition 1 is general, its application with your approach requires equation 8. On lines 160-161 the paper states:
> > "This formula implies an important fact that we only need the covariance of the teacher’s probability
> > to measure the “goodness” of a certain DA technique, no need for the student"
> > But this statement is only true if the assumption of a well-trained student is true. Which is not true in general, as your new accuracy data shows.
> > Now, it may be that the effect of this invalid assumption is minor, but you still need to (even empirically) demonstrate that this is the case in practice.

---

> > > ### Author Response · Authors · 2022-08-08
> > > **Thanks for following up!**
> > >
> > > > Even though Proposition 1 is general, its application with your approach requires equation 8. On lines 160-161 the paper states: "This formula implies an important fact that we only need the covariance of the teacher’s probability to measure the “goodness” of a certain DA technique, no need for the student" But this statement is only true if the assumption of a well-trained student is true. Which is not true in general, as your new accuracy data shows. Now, it may be that the effect of this invalid assumption is minor, but you still need to (even empirically) demonstrate that this is the case in practice.
> > >
> > > Thank you! We understand your point and actually agree with it. Currently, we honestly do not have more empirical evidence to show the well-trained student holds for *all* the pairs in practice (and to what degree can it be called "well"-trained? I do not have a rigorous answer either). Yet, we have the following comments that may help resolve your concern.
> > >
> > > - Let's go back to the key formula, Eq. 8, before the approximation. As mentioned previously, ideally, for each DA, we can calculate Eq. 8 with different students and have different DA orders. Presumably, some students may rank, say cutmix ahead of mixup, while some may rank cutmix behind mixup. Then, *which student should we trust when we want to have just one ranking of the DAs in practical applications*? This means, even if we do not have the approximation problem, we would still need to choose a certain "oracle" student to decide which DA is better.
> > > - With this as background, let's look at Eq. 8 again. **The approximation essentially says, we use the student that can perform exactly the same as (or very close to) the teacher, i.e., $\mathbf{p}^{(t)}(x) = \mathbf{f}(x)$, as the "oracle" student.** Then, the real question is, *does such a student exist in practice?* The answer is clearly YES -- we can easily find students that can mimic the teacher with a very small error (and actually, as is well known,  neural networks have the capacity to approximate any function with a desired precision).
> > >
> > > Hope this can help explain why our method still works well in practice. For the statement "*This formula implies an important fact that we only need the covariance of the teacher’s probability to measure the “goodness” of a certain DA technique, no need for the student*", we will add an explicit condition that we choose the student that can perform very closely to the teacher as the "oracle student" discussed above. This should make the statement more rigorous.
> > >
> > > **Again, thank you so much for being with us so far!** Let us know if you have more thoughts! Thanks a lot!

---

> > > > ### Comment · Reviewer_n9FN · 2022-08-08
> > > > **reply**
> > > >
> > > > Hi,
> > > >
> > > > This is not the OP reviewer but I wanted to share some of my own thoughts (which may or may not count for much). It would make sense to think of violating the 'well trained student' assumption as something that is more along a sliding scale, rather than a yes/no answer.
> > > >
> > > > I think this could have been supplemented with a nice scatterplot, where one plots on the x axis equation 8 and on the y axis its approximation. Points on the plot would comprise random (or all) enumerations of student/teacher pairs. I would then expect the correlation (e.g. $R^2$) between (x,y) to go down as the generalisation gap between the student and teacher increases. I don't know if you have any experiments where you intentionally make the student much worse than the teacher to see what happens, but that negative result would help with the figure, which in turn will help the story you're trying to tell.
> > > >
> > > > Thanks.

---

> > > > > ### Author Response · Authors · 2022-08-09
> > > > > **Great thanks for this suggestion!**
> > > > >
> > > > > Thank you so much for the suggestion! We are working on this. Hopefully, we can get some plots before the rebuttal ends.

---

> > > > > ### Author Response · Authors · 2022-08-09
> > > > > **Results update and more discussions**
> > > > >
> > > > > Dear reviewer Tm5Y and n9FN,
> > > > >
> > > > > Thank you *so much* for keeping giving feedback on our paper! And thank reviewer n9FN for giving us the suggestion. Following the suggestion, we train students and save checkpoints periodically over the training process, so we obtain multiple students, weak to strong. Then we use these students to estimate the *sum of covariance terms in the last line of Eq. 7*: $\frac{1}{N^2} \sum\_{1\le j <k \le N} \mathrm{Cov}[q(x_j), q(x_k)] $, which is our ultimate goal to estimate.
> > > > >
> > > > > Per the suggestion of reviewer n9FN, we estimate two versions of $\frac{1}{N^2} \sum\_{1\le j <k \le N} \mathrm{Cov}[q(x_j), q(x_k)] $:
> > > > >   1. $\mathrm{Cov1} = \frac{1}{N^2} \sum\_{1\le j <k \le N} \mathrm{Cov}[q(x_j), q(x_k)] $, where $q(x) = \mathbf{p}^{(t)}(x_i)^\top \log (\mathbf{f}(x_i))$
> > > > >   2. $\mathrm{Cov2} = \frac{1}{N^2} \sum\_{1\le j <k \le N} \mathrm{Cov}[q(x_j), q(x_k)] $, where $q(x) = \mathbf{p}^{(t)}(x_i)^\top \log ( \mathbf{p}^{(t)}(x_i))$ -- i.e., using the approximation in Eq. 8
> > > > >
> > > > > Results are presented below. Pair: VGG13/VGG8, Dataset: CIFAR100, DA: cutmix. We tried three students at different epochs. Format: $\mathrm{Cov1} / \mathrm {Cov2}$
> > > > >
> > > > > |  | Epoch 10 | Epoch 150 | Epoch 240 |
> > > > > |---|---|---|---|
> > > > > | Accuracy | 49.15% | 58.66% | 73.55% |
> > > > > | Run #1 | 0.0000031704 / 0.0000036407 | 0.0000033388 / 0.0000024198 | 0.0000019557 / 0.0000027502 |
> > > > > | Run #2 | 0.0000016844 / 0.0000024425 | 0.0000017658 / 0.0000029721 | 0.0000016558 / 0.0000032665 |
> > > > > | Run #3 | 0.0000028793 / 0.0000022399 | 0.0000014990 / 0.0000029898 | 0.0000024180 / 0.0000020152|
> > > > >
> > > > > Honestly, we do see obvious patterns here. **The foremost problem, as you may notice, is that the multiple-run results vary quite a lot** (note Run #1/#2/#3 of each column are using *exactly the same* script). This literally means $\mathrm{Cov1}$ and  $\mathrm {Cov2}$ are very challenging to estimate (and our presented results are unreliable since the variation is too large). We can actually analyze to see how challenging it is. Take $\mathrm{Cov1}$ as example:
> > > > > - Essentially, it is a sum of $\frac{N*(N-1)}{2}$ covariance of paired RVs. Unfornaturely, $N$ is very large here -- $N$ is the sample size of the sampled training set. In the case of CIFAR100, $N=50,000$. That makes $\frac{N*(N-1)}{2}$ in the order-of-magnitude of $10^9$. Namely, **we mean to estimate the sum of covariance of $\sim 10^9$ pairs of RVs**. This is very challenging if we estimate it by sampling.
> > > > >
> > > > > So to reviewer n9FN first, sorry we cannot have the plots you suggested before the rebuttal ends because we do not have a reliable estimation of the covariance terms at present. (This said, we cannot eliminate the possibility that we did not do the estimation the best way due to the limited time. We shall keep trying your idea later. Thank you!)
> > > > >
> > > > > ---
> > > > > **Then, how should we properly look at the "well-trained student" assumption?**
> > > > >
> > > > > As we analyzed above, ideally, given a set of DA schemes, their "goodness" order is decided **with the student in the loop** (note Eq. 7 has the student $\mathbf{f}(x)$ as a part) -- this is also more in line with our intuition (as reviewer Tm5Y mentioned in the original comments: "*one cannot just ignore the student in determining the effectiveness of the DA method*"). Consequently, different students (not only different networks but also the same network at different training stages) will give different DA orders in "their" opinions.
> > > > >
> > > > > For practical use, we prefer just one DA order. This means, we *must* choose a student to decide the DA order. Since anyway we will choose a student, choosing a "well-trained" student is naturally more reasonable and pertinent to our goal (and such a student *does* exist in practice as we analyzed previously).  This is how the "well-trained student" assumption is brought up.  Results turn out to show it works pretty well *across different students*.
> > > > >
> > > > > We will add these discussions to the manuscript as a background for the following statement
> > > > > > This formula implies an important fact that we only need the covariance of the teacher’s probability to measure the “goodness” of a certain DA technique, no need for the student"
> > > > >
> > > > > It should be
> > > > > > To make a metric for practical use, we have the *freedom* to decide which student is used to estimate the covariance term in Eq. 7. Then we *intentionally choose* the student that performs the same as (or very close to) the teacher to further reduce the covariance term in Eq. 7, which only involves the teacher, no need for the student. Despite not involving the student terms, the metric works pretty well in capturing the generalization error of the students trained with different DA schemes.
> > > > >
> > > > > We believe this is the most we can do at this moment (since it's very close to the rebuttal end). **Thank you *so much* for your awesome suggestions all along the way! We are lucky to have the reviewers like you! Tremendous thanks!**

---

### Official Review · Reviewer_n9FN · 2022-07-10

**Rating:** 7
**Confidence:** 3
**Soundness:** 3 good
**Presentation:** 2 fair
**Contribution:** 3 good

**Summary:**

The paper establishes a connection between data augmentation (DA) and generalisation of student-teacher training, namely showing theoretically that given two subsequences of the original dataset D (where the subsequences differ by some data augmentation scheme), the subsequence that performs better is the one that induces less variation in the predicted probabilities of the teacher network.

**Edited review**

In light of the authors _active discussion and contributions_ during this phase, as well as their addressal of my own concerns, I will tentatively raise my score from 4 to 7. Obviously this is conditional on some changes to the manuscript, namely:
- Making the math more digestible (we already discussed this in great detail)
- Preferring plots to large tables where applicable, or having plots supplement them. I leave this to the authors' discretion since space constraints will mean decisions will have to be made as to what should be a table vs what should be a plot, and what should be relegated to the appendix.
- Further work on plotting Equation 8 vs its approximation vs both 'bad' and 'good' students. The authors have acknowledged this and even provided preliminary results.

Thanks, and good work.

**Questions:**

My questions mainly lie in the aforementioned weaknesses.

**Strengths And Weaknesses:**

# Strengths

- The efficacy of any DA scheme for student training could in principle be evaluated before the training of any student model, since computing the variance term $V[\Delta]$ only requires the pretrained teacher model.

- Experiments support the proposed theory, and are numerous, run over many different architectural variants as well as data augmentation schemes.

- Proposal of an entropy-based scheme to improve results.

# Weaknesses

Some of the notation in Section 3.1 and 3.2 is incredibly confusing, and looks to be rushed. Firstly, I would recommend simply using $\log$ instead of $\mathcal{L}$ to denote the logarithm. Alternatively, if you want to be more abstract, you can simply define the cross-entropy loss as $\mathcal{L}_{CE}(x, y; f) = y^T f(x)$ and it would make equation (6) more readable. In fact, you have already defined it as such in Equation (1).

In Section 3.2, I have several concerns. In Equation (6), in line 1 you have an expectation $S \sim D^N$, yet in line 2 it confusingly turns into an expectation $x \sim D$, yet you still have the inner summation. Why isn't it just this instead:

$E_{x \sim D} [ p(x) \log f(x) ] + \text{const}$

i.e. the expectation implies the inner summation + multiplication over $1/N$, so having both seems redundant (and incorrect). Furthermore, in the inner multiplication you're missing the transpose, so it should be $p(x_n)^{T} \log f(x_n)$, since that is how it is defined in equation (4).

In Equation (7), I am also not sure why the last line expands out into a variance + covariance term. Covariance to me implies that you're computing the variance between two sets of random variables; don't we only have one random variable? (That random variable being $\Delta$, the empirical risk over whatever random draw $S \sim D$.) Maybe I am missing something crucial, but it seems like one could just remove the last line of Equation (7), leaving us with:

$V_x[ p(x)^T \log f(x) ]$,

then say that the variance is going to be proportional to the $p^{(t)}$ and that $f$ is held constant. Actually, it seems like from Equation (9) that the covariance is over the $p$'s for the different classes. What is confusing is that, for a given $x$, you are indexing into its predicted probability for class $i$ as as $p^{(t)}(x_i)$, and $x_i$ is apparently the $i$'th example as you have implied in equations (3) and (4). If $p^{(t)}$ outputs a probability distribution over x, then you should be using $p^{(t)}(x)_i$ instead.

"Each element in S is drawn the same underlying distribution D." - I don't follow this, since it says earlier that $S_1$ and $S_2$ are elements from $D$ but not necessarily iid. If so, how can $E_{x \sim S}$ turn into $E_{x \sim D}$?

Table 1: I appreciate that the results are averages over three runs, but please indicate the standard deviation. Since space seems a little tight in this table, maybe just indicate the stdevs below each number in parentheses. Also such information can be nicely condensed into a scatterplot with one axis denoting test accuracy and the other denoting mean stdev. When necessary, prefer plots over big tables, and leave big tables to the appendix.

---

> ### Author Response · Authors · 2022-08-02
> **Responses to Reviewer n9FN (Part 2)**
>
> `Q5`: then say that the variance is going to be proportional to the  $p^{(t)}$ and that $\mathbf{f}$ is held constant. Actually, it seems like from Equation (9) that the covariance is over the $p$'s for the different classes. What is confusing is that, for a given $x$, you are indexing into its predicted probability for class $i$ as as $p^{(t)}(x_i)$ , and $x_i$ is apparently the $i$'th example as you have implied in equations (3) and (4). If $p^{(t)}$ outputs a probability distribution over x, then you should be using $p^{(t)}(x)_i$  instead.
>
> `A5`: *it seems like from Equation (9) that the covariance is over the $p$'s for the different classes* -- This might be misreading. The covariance is *always* for different examples, not for different classes.
>
> `Q6`: "Each element in S is drawn the same underlying distribution D." - I don't follow this, since it says earlier that $S_1$ and $S_2$ are elements from $D$ but not necessarily iid. If so, how can $E_{x \sim S}$  turn into $E_{x \sim D}$ ?
>
> `A6`: Thanks to your help, we update the derivation of Eq. 6. See above `A2`.
>
> `Q7`: Table 1: I appreciate that the results are averages over three runs, but please indicate the standard deviation. Since space seems a little tight in this table, maybe just indicate the stdevs below each number in parentheses. Also such information can be nicely condensed into a scatterplot with one axis denoting test accuracy and the other denoting mean stdev. When necessary, prefer plots over big tables, and leave big tables to the appendix.
>
> `A7`: We report the std in Tab. 1 as follows. Meanwhile, thanks for the suggestion! We will use plots and move the big tables (with std reported) to Appendix.
>
> |  | WRN-16-2 | VGG8 | Res20 | ShuffleV2 | VGG8 | MobileV2 |
> |---|---|---|---|---|---|---|
> | KD+Identity | 0.0136 | 0.0065 | 0.0081 | 0.0137 | 0.0196 | 0.0366 |
> | KD+Flip | 0.0101 | 0.0060 | 0.0060 | 0.0072 | 0.0105 | 0.0076 |
> | KD+Flip+Crop | 0.0097 | 0.0182 | 0.0092 | 0.0062 | 0.0059 | 0.0149 |
> | KD+Cutout | 0.0090 | 0.0156 | 0.0115 | 0.0101 | 0.0055 | 0.0104 |
> | KD+AutoAugment | 0.0290 | 0.0016 | 0.0041 | 0.0099 | 0.0213 | 0.0440 |
> | KD+Mixup | 0.0143 | 0.0044 | 0.0121 | 0.0060 | 0.0176 | 0.0044 |
> | KD+CutMix | 0.0108 | 0.0145 | 0.0034 | 0.0107 | 0.0042 | 0.0178 |
> | KD+CutMixPick (S ent.) | 0.0118 | 0.0111 | 0.0095 | 0.0070 | 0.0249 | 0.0064 |
> | KD+CutMixPick (T ent.) | 0.0095 | 0.0061 | 0.0120 | 0.0148 | 0.0115 | 0.0233 |
>
>
> **Lastly, thank you so much for helping us improve the paper so far! Please let us know *asap* if you have any further questions. We are actively available during this rebuttal!**

---

> ### Author Response · Authors · 2022-08-02
> **Responses to Reviewer n9FN (Part 1)**
>
> Thank you so much for your constructive comments! We address your concerns as follows.
>
> `Q1`: Firstly, I would recommend simply using $\log$  instead of $\mathcal{L}$ to denote the logarithm.
>
> `A1`: We will change all the logarithm notation to $\log$ as you suggested. Thank you!
>
> `Q2`: In Equation (6), in line 1 you have an expectation $S \sim D^N$, yet in line 2 it confusingly turns into an expectation $x \sim D$ , yet you still have the inner summation. Why isn't it just this instead:
> $$
> E_{x\sim D} [p(x)\log f(x)] + \text{const}
> $$
> i.e. the expectation implies the inner summation + multiplication over $1/N$, so having both seems redundant (and incorrect). Furthermore, in the inner multiplication you're missing the transpose, so it should be $p(x_n)^{T} \log f(x_n)$, since that is how it is defined in equation (4).
>
> `A2`: First, your comment about the transpose is correct. We'll add the missing transpose. Thanks!
>
> Next, we walk step by step from $S \sim D^N$ to $x \sim D$:
>
> $$
> \begin{aligned}
> E\_{S \sim D^N} [ \Delta ] &= E\_{S \sim D^N} [ \hat{R}_S(\mathbf{f})] + \text{Const} \\\\
> & = E\_{ \\{x_1,x_2,...,x_N \\} \sim D^N} [ \frac{1}{N} \sum\_{n \in [N]} p^{(t)}(x\_n)^\top  \log(\mathbf{f}(x\_n)) ] + \text{Const} \\\\
> &= \frac{1}{N} E\_{ \\{x_1,x_2,...,x_N \\} \sim D^N} [p^{(t)}(x\_1)^\top  \log(\mathbf{f}(x\_1)) + ... + p^{(t)}(x\_N)^\top  \log(\mathbf{f}(x\_N))] + \text{Const} \\\\
> &= \frac{1}{N} \big( E\_{ \\{x_1,x_2,...,x_N \\} \sim D^N} [p^{(t)}(x\_1)^\top  \log(\mathbf{f}(x\_1))] + ... + E\_{ \\{x_1,x_2,...,x_N \\} \sim D^N} [p^{(t)}(x\_N)^\top  \log(\mathbf{f}(x\_N))] \big) + \text{Const} \\\\
> & = \frac{1}{N} \big( E\_{ x_1 \sim D} [p^{(t)}(x\_1)^\top  \log(\mathbf{f}(x\_1))] + ... + E\_{ x_N \sim D} [p^{(t)}(x\_N)^\top  \log(\mathbf{f}(x\_N))] \big) +  \text{Const} \\\\
> & = \frac{1}{N} \cdot N \cdot E\_{x \sim D}  [p^{(t)}(x)^\top  \log(\mathbf{f}(x))] + \text{Const} \\\\
> & = E\_{x \sim D}  [p^{(t)}(x)^\top \log(\mathbf{f}(x))] + \text{Const}
> \end{aligned}
> $$
>
> So, yes! Your comment is correct! We have an extra summarization inside. Thank you *so much* for discovering this! We'll proofread the paper more carefully to eliminate such mistakes. Meanwhile, notably, **with the new equation above, it is even more clear to see that for $S_1$ and $S_2$, the $ E\_{S \sim D^N} [ \Delta ] $ is the same. Namely, our conclusion in Line 146 still holds.**
>
>  `Q3`: In Equation (7), I am also not sure why the last line expands out into a variance + covariance term. Covariance to me implies that you're computing the variance between two sets of random variables; don't we only have one random variable? (That random variable being $\Delta$, the empirical risk over whatever random draw $S \sim D$.)
>
> `A3`: We have many random variables. E.g., each sample $x$ in $S$ is taken as a random variable. And *the covariance is exactly describing the dependency among different samples* (in the form of $p^{(t)}(x) \log(\mathbf{f}(x))$).
>
> `Q4`: Maybe I am missing something crucial, but it seems like one could just remove the last line of Equation (7), leaving us with:
> $$
> \mathrm{Var}\_x [p(x)^\top \log(\mathbf{f}(x))]
> $$
>
> `A4`: We explain the derivation of Eq. 7 step by step as follows:
>
> **[Edited 08/04]** Following your suggestion, we revise the following derivation to make it clearer. First, we define $q(x_i) = p^{(t)}(x_i)^\top \log (\mathbf{f}(x_i))$, and use $\mathrm{Var}[\cdot]$ for the variance of a random variable, $\mathrm{Cov}[\cdot, \cdot]$ for covariance. Then,
> $$
> \begin{aligned}
> \mathrm{Var}\_S [\hat{R}\_S({\mathbf{f}})] &= \mathrm{Var}\_S [ \frac{1}{N} \sum\_{i=1}^N  q(x\_i) ] \\\\
> &= \frac{1}{N^2} \mathrm{Var}\_S [ \sum\_{i=1}^N  q(x\_i) ] \\\\
> &= \frac{1}{N^2} \mathrm{Cov}\_S [ \sum_{j=1}^N q(x_j), \sum_{k=1}^N q(x_k) ] \\\\
> &= \frac{1}{N^2} \big( \sum\_{i=1}^N \mathrm{Var}\_{x_i} [ q(x\_i) ] + 2 \sum\_{1\le j<k \le N} \mathrm{Cov}[ q(x_j), q(x_k) ] \big) \\\\
> &= \frac{1}{N^2} \big( N \cdot \mathrm{Var}\_{x} [ q(x) ] + 2 \sum\_{1\le j<k \le N} \mathrm{Cov}[ q(x_j), q(x_k) ] \big) \\\\
> & = \frac{1}{N} \mathrm{Var}\_x[q(x)] + \frac{2}{N^2} \sum\_{1\le j<k \le N} \mathrm{Cov}[ q(x_j), q(x_k) ]
> \end{aligned}
> $$

---

> > ### Comment · Reviewer_n9FN · 2022-08-04
> > **Reply**
> >
> > Hi, thanks for addressing my questions. I am satisfied with the re-derivation of the expectation (over S) for $\Delta$.
> >
> > For the variance term, I missed that you are using a particular identity which is the variance for a sum of random variables. It appears however that for `A4` you are missing the summation behind $C_{x_i, x_j}$ in the last two lines. I still would like this derivation to be slightly more clear using this identity [1]. Maybe you can combine the following with the explained derivation you have given me:
> >
> > $$
> > \begin{align} \text{Var}[ \sum_{k=1}^{n} x_k ] & = \text{Cov}[ \sum_{j=1}^{n} x_j, \sum_{k=1}^{n} x_k] \\\\
> > & = \sum_{j=1}^{N} \text{Var}[x_j] + 2\sum_{j < k} \text{Cov}[x_j, x_k]
> > \end{align},$$
> >
> > where $x_j$ is basically the $p^{(t)}(x_j) \log f(x_j)$ random variable term (abusing notation a little here since I'm re-using x).
> >
> > (1) https://math.stackexchange.com/questions/2413664/different-rules-for-calculating-the-variance-of-a-sum?rq=1
> >
> > I am happy to increase my score after both derivations have been made clearer in the revised manuscript, but the degree to which it increases will also depend on issues and points raised with the other reviewers. Thanks.

---

> > > ### Author Response · Authors · 2022-08-04
> > > **Thank you for the further comments!**
> > >
> > > Thank you so much for following up with us!
> > >
> > > We have revised the derivation above to make it clearer following your suggestions. The manuscript will be updated very soon **[ Edited 08/05: the manuscript has been updated ]**. Thank you so much for helping us improve the paper and considering raising the score! *If you have any further questions, let us know any time!*

---

> > > ### Author Response · Authors · 2022-08-09
> > > **Lastly, thank you!**
> > >
> > > Dear Reviewer n9FN,
> > >
> > > Thank you for pointing out the derivation mistakes in our original manuscript. Also, thank you for giving suggestions voluntarily to us when we address the concern from reviewer Tm5Y. Your comments help *a lot* to improve our paper!
> > >
> > > Since your major concern is about the formulas, which we have corrected thanks to your awesome detailed comments, we were wondering, could you please generously consider raising your score *even a little bit* as a recognition of our efforts so far? Thank you *so much*!
> > >
> > > Sincerely,
> > >
> > > Authors

---

> > > > ### Comment · Reviewer_n9FN · 2022-08-09
> > > > **Edited review**
> > > >
> > > > **Edited review**
> > > >
> > > > In light of the authors _active discussion and contributions_ during this phase, as well as their addressal of my own concerns, I will tentatively raise my score from 4 to 7. Obviously this is conditional on some changes to the manuscript, namely:
> > > > - Making the math more digestible (we already discussed this in great detail)
> > > > - Preferring plots to large tables where applicable, or having plots supplement them. I leave this to the authors' discretion since space constraints will mean decisions will have to be made as to what should be a table vs what should be a plot, and what should be relegated to the appendix.
> > > > - Further work on plotting Equation 8 vs its approximation vs both 'bad' and 'good' students. The authors have acknowledged this and even provided preliminary results.
> > > >
> > > > Thanks.

---

> > > > > ### Author Response · Authors · 2022-08-09
> > > > > **Thank you!**
> > > > >
> > > > > Thank you *so much* for generously raising the score! Your suggestions are well-taken. We *promise* to materialize the 3 conditional changes in our revised version. Thanks again!

---

### Official Review · Reviewer_JbXW · 2022-07-10

**Rating:** 5
**Confidence:** 4
**Soundness:** 3 good
**Presentation:** 3 good
**Contribution:** 2 fair

**Summary:**

Knowledge distillation (KD) is to improve a low-capacity student network's performance with a high-capacity teacher network. Current works focus on how to define KD loss. This work proposes to improve KD from data augmentation (DA). This work proposes an entropy-based data-mixing DA to improve two KD methods: KD and CRD.

**Questions:**

The main concern is the novelty of the proposed methods (see weakness).

**Ethics Review Area:**

["I don’t know"]

**Limitations:**

The authors have addressed the the limitations and potential negative societal impact of their work.

**Strengths And Weaknesses:**

Strengths:
1. The method is simple and clearly presented.
2. The experiments are done on different data augmentation techniques.
3. Results on CIFAR-100 and TinyImageNet show increased performance.

Weaknesses:
There are three contributions in the method part: (1) data augmentation (2) longer training and (3) cutmixpick. (1) and (2) are already investigated in works [1][2]
[1] Cui, Wanyun, and Sen Yan. "Isotonic Data Augmentation for Knowledge Distillation." IJCAI2021
[2] Beyer, Lucas, et al. "Knowledge distillation: A good teacher is patient and consistent." CVPR2022.
The improvement from (3) is very minimal.


After rebuttal:
The authors have addressed my concerns. I increased my score to boardline accept after considering other reviewrs' comments.

---

> ### Author Response · Authors · 2022-08-02
> **Responses to Reviewer JbXW**
>
> Thank you so much for your constructive comments! We address your concerns as follows.
>
> `Q1`: Weaknesses: There are three contributions in the method part: (1) data augmentation (2) longer training and (3) cutmixpick. (1) and (2) are already investigated in works [1][2] [1] Cui, Wanyun, and Sen Yan. "Isotonic Data Augmentation for Knowledge Distillation." IJCAI2021 [2] Beyer, Lucas, et al. "Knowledge distillation: A good teacher is patient and consistent." CVPR2022. The improvement from (3) is very minimal.
>
> `A1`: We agree that [1] and [2] are related to our works since they also study the KD problem and are relevant to DA. We will cite these two papers and discuss them in our related work section.
>
> Yet it is worth noting that our work is *distinct* from [1] and [2].  Reviewer JbXW summarizes the three contributions in our method part (i.e., Sec. 4). Yet, this summarization might *miss the major point* of our paper.
> - Although the proposed CutMix-Pick algorithm in Sec. 4 is one claimed contribution, it is *not the major one* -- see our summarized contributions in Line 65 - 74.
> - Notably, our paper title, core theory part (Sec. 3), summarized contribution (Line 65 - 74), and conclusion (Line 336 - 342), *all* highlight the major point/contribution of this paper: what defines a “good” DA in KD and we attempt to have a theoretical and precise answer to this scientific question. **The derived answer (Proposition 1 and metric in Eq. 10)  is what we claimed as the major contribution (see our summarized contributions in Line 65 - 74)**. In this regard, *none of the prior works have answered this question in a theoretical manner*, including [1] and [2]. Thus, we believe our paper is novel.
> - Reviewer JbXW summarizes “*longer training*” as a contribution to our method, which might be a misreading. Note the longer training is introduced in Line 298 - Line 303, which is in our “Experimental Results” section, not the methodology part (Sec. 3 and 4). Clearly, we never consider it as a "contribution" from this paper. We have the longer training experiments simply to show how the proposed theory can be used for higher performance in practice, as an extra bonus. Even without the “longer training”, note the paper is self-contained: theory proved (Proposition 1) and empirically validated (Tab. 1 and 2).
>
> **Lastly, thank you so much for helping us improve the paper so far! Please let us know *asap* if you have any further questions. We are actively available during this rebuttal!**

---

### Author Response · Authors · 2022-08-05
**Sincerely looking forward to your further feedback!**

Dear reviewers,

Thank you *so much* for helping improve our paper so far!

The author-reviewer discussion period has passed around half. We are looking forward to your further comments. If you prefer to check on them after this weekend (08/06, 08/07), it is perfectly fine! Meanwhile, **we still hope you may take a quick look at least (like 5 minutes), in case you may expect more experiments/analyses/clarifications, so that we can finish them over this weekend.**

Thank you so much for being with us so far! Have a wonderful weekend!

--- Authors, 08/05

---

### Meta-Review · Area_Chair_YkKh · 2022-08-26

**Recommendation:** Accept
**Confidence:** Certain

**Metareview:**

After a lively and interactive author discussion period all reviewers ended up recommending to accept this paper.
The work examines the ways in which different data augmentation schemes can increase knowledge distillation performance, providing some theoretical analysis with actionable insights and experiments to back it up. The work focuses on the generalization gap of the student under different sampling schemes, and asserts that their study leads to the conclusion that a good data augmentation scheme should reduce the variance of the empirical distilled risk between the teacher and student. Reviewers were generally positive about the clarity of the manuscript after some changes during the author discussion.

The AC recommends acceptance.

**Award:**

No

---

### Decision · Program_Chairs · 2022-09-14

Accept